# Temporal stability of stimulus representation increases along rodent visual cortical hierarchies

Eugenio Piasini [1,7], Liviu Soltuzu[2,6,7], Paolo Muratore [2], Riccardo Caramellino[2], Kasper Vinken [3,4], Hans Op de Beeck[5], Vijay Balasubramanian[1] & Davide Zoccolan [2✉]

Cortical representations of brief, static stimuli become more invariant to identity-preserving transformations along the ventral stream. Likewise, increased invariance along the visual hierarchy should imply greater temporal persistence of temporally structured dynamic stimuli, possibly complemented by temporal broadening of neuronal receptive fields. However, such stimuli could engage adaptive and predictive processes, whose impact on neural coding dynamics is unknown. By probing the rat analog of the ventral stream with movies, we uncovered a hierarchy of temporal scales, with deeper areas encoding visual information more persistently. Furthermore, the impact of intrinsic dynamics on the stability of stimulus representations grew gradually along the hierarchy. A database of recordings from mouse showed similar trends, additionally revealing dependencies on the behavioral state. Overall, these findings show that visual representations become progressively more stable along rodent visual processing hierarchies, with an important contribution provided by intrinsic processing.

[1] Computational Neuroscience Initiative, University of Pennsylvania, Philadelphia, PA, United States. [2] Visual Neuroscience Lab, International School for Advanced Studies (SISSA), Trieste, Italy. [3] Boston Children's Hospital, Harvard Medical School, Boston, MA, USA. [4] Laboratory for Neuro- and Psychophysiology, Department of Neurosciences, KU Leuven, Leuven, Belgium. [5] Department of Brain and Cognition, Leuven Brain Institute, KU Leuven, Leuven, Belgium. [6]Present address: Blue Brain Project, École polytechnique fédérale de Lausanne (EPFL), Campus Biotech, Geneva, Switzerland. [7]These authors contributed equally: Eugenio Piasini, Liviu Soltuzu. ✉email: zoccolan@sissa.it

The visual system of mammals supports discrimination and categorization of objects despite variation in their appearance, resulting from, e.g., translations, rotations, and changes of scale[1–3]. Decades of research on nonhuman primates have shown that this ability (known as invariant object recognition) results from processing in the ventral visual stream, a hierarchy of cortical areas along which invariance to object transformations emerges progressively from largely feedforward computations[1,4,5]. While neurons at early stages of this pathway (e.g., simple cells in primary visual cortex; V1) are highly sensitive to variation in the appearance of their preferred stimuli (e.g., the position of oriented edges), units at the apex of the hierarchy (the anterior inferotemporal cortex; IT) maintain their tuning for visual objects despite, e.g., position and scale changes. Thus, IT neurons better support invariant object recognition as compared to lower-level areas[6–10]. However, these conclusions mainly derive from studies relying on presentation of brief, static visual stimuli, designed to engage feedforward processing while minimizing the influence of recurrent and top–down processes and adaptation. Thus, it remains unknown how invariance along the ventral stream emerges under naturalistic viewing conditions involving dynamic stimuli (e.g., natural movies lasting several seconds).

Under naturalistic stimulation, an increase of invariance along the ventral stream (as tested with static objects) should translate into an increase of temporal stability of neuronal responses, possibly complemented by increases of stability due to broader temporal tuning of neurons in higher areas[11,12]. The link between invariance and temporal stability can be understood by considering an idealized invariant object detector (Fig. 1). Such a unit would respond to its preferred object (e.g., a "rat head") within its receptive field (RF; Fig. 1a, yellow shape) regardless of the position, size or orientation. Thus, the unit should start firing to report the presence of the object when it enters its RF, and should keep firing at roughly the same rate while the object smoothly transforms (e.g., translates/rotates) within the RF (Fig. 1b top, green trace). By contrast, a poorly invariant, lower-level unit (e.g., a V1-like edge detector) should fire transiently (Fig. 1b bottom, green trace), given that features matching its selectivity will quickly enter and exit its much smaller RF (orange circle in Fig. 1a), as the movie unfolds. The relationship between invariance and temporal stability (or slowness) is at the base of many

theoretical accounts of invariance along the ventral stream, where temporal continuity of the visual input is exploited by unsupervised learning mechanisms to build invariant representations[13–22]. This relation between invariance and temporal stability is also consistent with any feedforward processing hierarchy (e.g., convolutional neural networks[23]), in which the instantaneous response of units to the current input only depends on the input itself and not on previous activity of the unit or the current state of the network. In this scenario, an object smoothly transforming within a dynamic scene is treated and processed as a sequence of static, independent snapshots, so that object invariance translates into temporal stability. Of course, increasing temporal stability can also arise from other mechanisms, such as a broader temporal tuning of neurons in higher areas. In this case, a hierarchical grow of temporal scales would be observable independently of whether object identity remains stable while the dynamic stimulus unfolds over time.

An alternative scenario is one where nontrivial history- and state-dependent computations significantly modify the temporal dynamics of neuronal responses, when tested with naturalistic dynamic inputs. For instance, adaptation has been shown to depress visual cortical responses to repeated presentation of perceptually similar visual stimuli in both monkeys and humans[24–26], resulting in reduced discriminability of the adapted stimuli[27]. fMRI evidence in humans suggests that the impact of adaptation mechanisms may increase along the ventral stream[28–30] and that neural responses in higher areas of the hierarchy may be driven more strongly by transient than sustained stimuli[30,31]. Similarly, in rats, adaptation has been shown to increase in magnitude along the cortical shape-processing hierarchy[32], attenuating the responses to predictable stimuli[33]. Finally, in primates, adaptation is at least partially preserved across object transformations (e.g., a smaller object can adapt the response to a larger object), as shown in both neurophysiological[34–36] and behavioral studies[37,38]. Together, these effects could counteract increases in response persistence from increased invariance or broader temporal tuning, as shapes that are temporally stable (e.g., a translating or expanding object) could strongly and continuously adapt neuronal responses until a new, surprising stimulus (i.e., a novel object) enters the neurons' RFs. This is illustrated in Fig. 1b (blue curves), where the idealized "rat" detector unit, after an initial, transient response to its preferred object (the "rat head"), stops reporting the presence of the object within its RF, thus behaving similarly to a lower-level, edge-detector unit. More broadly, the predictive coding framework posits that, within certain cortical circuits, only error or surprise signals are encoded, their temporal dynamics naturally depending on a top–down signal carrying the input prediction[39–41]. This leads to an alternative hypothesis that the timescale and persistence of neural responses do not increase across the cortical hierarchy, as each area encodes surprising features of the response of the previous area (example cartoon in Fig. 1a, b). Some of these intuitions can be formalized by a computational model of adaptation[42], such that response timescales to dynamic stimuli become progressively shorter as a function of the strength of adaptation (see Supplementary Text).

More generally, mechanisms that alter the current state of the visual pathway in an activity-dependent way can have other types of effect on the temporal dynamics of the neural code, regardless of whether they tend to make representations more temporally sparse (like adaptation) or more stable (like temporally extended integration of synaptic stimuli or recurrent excitation within local circuits[11,43]). In presence of noise, the existence of such mechanisms can be revealed by measuring the temporal span over which fluctuations of the firing rate are correlated (e.g., across repeated presentations of the same sensory input). This is typically called the intrinsic timescale of the recorded activity[44].

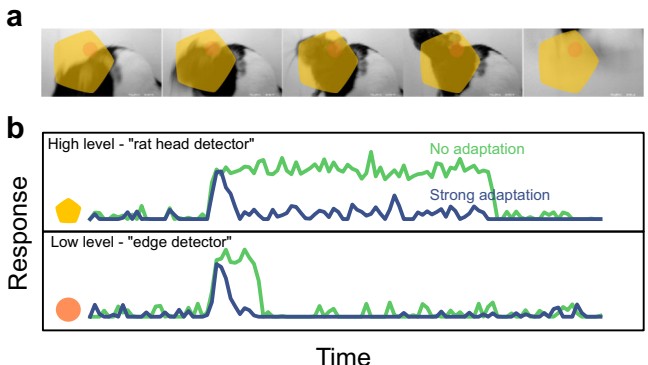

**Fig. 1 Effect of adaptation on the response timescale of high- vs. low-level feature detectors (cartoon). a** Dynamical visual stimulus (movie frames). Orange dot, yellow shape: idealized receptive field of a low-level feature detector neuron ("edge detector," orange) and a high-level feature detector ("rat head detector," yellow). **b** Single-trial response of the two example neurons, when adaptation is absent (green trace) and when adaptation is strong (blue trace). Note how adaptation shortens the timescale of the response.

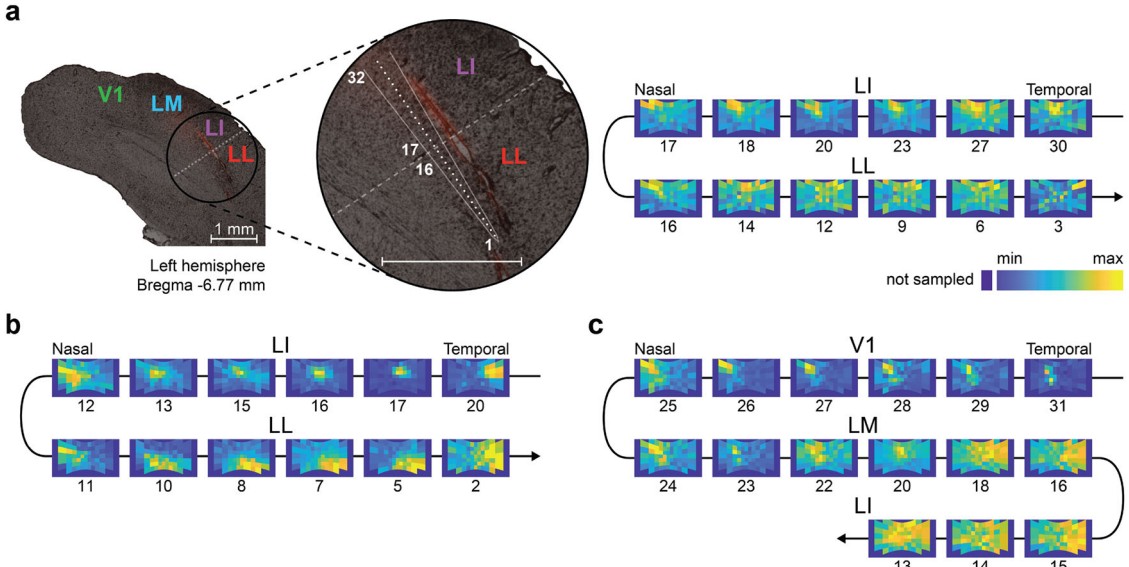

**Fig. 2 Functional identification of rat visual cortical areas. a** Left: overlay of a fluorescence image on a bright-field image of a coronal slice of rat visual cortex, from which neuronal recordings were obtained using a single-shank silicon probe with 32 recording sites, spanning 1550 $\mu$m. Before being inserted obliquely into cortex (to target lateral extrastriate areas LI and LL), the probe was coated with a red fluorescent dye that allowed reconstructing its insertion track (red stain). The close-up view shows the geometry of the probe (white lines) alongside the insertion track and the relative positions of the recording sites (white dots), from the tip (site # 1) to the base (site # 32) of the probe. Right: firing intensity maps showing the RFs of the units recorded at selected recording sites along the probe (indicated by the numbers under the RF maps). The reversal in the progression of the retinotopy between sites 16 and 17 marks the boundary between areas LI and LL (shown by a dashed line on the left panel). **b, c** Retinotopic progressions of the RFs recorded in two other example sessions. One session featured a single reversal between LL and LI within a span of the recording sites of 1550 $\mu$m (**b**). The other session featured two reversals, one between V1 and LM and another one between LM and LI, which were made possible by the larger depth spanned by the probe (3100 $\mu$m).

Interestingly, it was found that intrinsic timescales increase along various cortical hierarchies in primates and rodents[44,45]. However, the interaction between invariant encoding of object information and intrinsic processing along the ventral stream is not understood, and we do not know whether the net result is an ordered progression of temporal scales of neural processing. In addition, it is unknown how these processes interact with top–down modulation signals that reflect the attentive, motivational, or locomotory state of a subject[46–51] and whose influence on the processing of dynamic visual stimuli is still largely unexplored.

To address these questions, we must perform neurophysiological recordings along an object-processing hierarchy during presentation of dynamic stimuli, ideally from the bottom to the top stage of the processing chain. The primate ventral stream would be the ideal target for such investigation, but the anatomy and size of the monkey brain makes it difficult to compare more than a pair of visual cortical areas in a single study (e.g., see refs. [7,8,52]). By contrast, the small size of the rodent brain, the spatial contiguity of rodent visual cortical areas, and the possibility of using a much larger number of animals than in monkey studies make it easier to probe and compare multiple visual regions (e.g., see refs. [53–56]). Critically, during the last decade, a number of functional and anatomical studies in mice and rats have convincingly shown that visual information is processed in a hierarchical fashion across the many extrastriate visual areas that, in rodents, surround V1[53,54]. In particular, it has been shown that, along the progression of visual areas that run laterally to rat V1, many key tuning and coding properties evolve according to what is expected for a ventral-like, object-processing pathway[32,33,55–57], including the ability to support invariant visual object recognition—a finding that has been recently replicated in mice[58].

Based on this wealth of evidence, we decided to compare the timescales of cortical processing of dynamic visual stimuli across the rat analog of the ventral stream—i.e., V1 and the progression of extrastriate areas located laterally to V1: the lateromedial (LM), laterointermediate (LI), and laterolateral (LL) areas. To maximize the stability and duration of the recording sessions, as well as the number of repeated presentations of the movies used to probe these areas, neuronal recordings were performed under anesthesia[55,57,59]. We found that neural activity displays a hierarchy of temporal scales, as expected for a processing pathway where invariance is built through a cascade of feedforward computations. At the same time, we found that intrinsic temporal scales also become longer, and intrinsic correlations become more important in determining the overall temporal persistency of the neural representations, as one progresses from primary visual cortex (V1) toward higher-level areas. We checked the generality of these findings by analyzing two additional, existing datasets where visual cortex of awake rodents was probed with natural movies[12,60]. Interestingly, the trends found in our experiments were replicated (and even sharper) in awake, head-fixed mice that were running on a wheel, but they were attenuated during epochs in which the animals remained still, and were fully absent in awake, head-fixed rats that were body-restrained.

## Results

To investigate the temporal structure of neural representations of dynamic visual scenes, we used 32-channel silicon probes to record from cortical areas V1, LM, LI, and LL in anesthetized rats. These areas are arranged in an anatomical progression that allowed us to reach each sequentially, with a single diagonal penetration, stopping at a desired location, so as to distribute the recording sites across two or three adjacent regions (example session in Fig. 2a left, where the probe targeted LI and LL).

To infer the cortical area of each sampled unit, we used a functional identification procedure that is standard in electrophysiology and imaging studies of rodent visual cortex[53,54]. That is, we tracked the progression of the RFs recorded along the probe, mapping reversals of retinotopy that delineate borders between adjacent areas[55–57]. To this end, before the main stimulus, we run an RF mapping procedure, by displaying drifting bars with four different orientations over a grid of visual field locations. Figure 2a (right) shows the outcome for an example session, where the RFs recorded along the linear probe shown on the left are displayed as firing intensity maps over the visual field spanned by the monitor. Tracking this progression from the base (channel 32) to the tip (channel 1) of the probe revealed a reversal of the retinotopy (channels 16–17) corresponding to the boundary between areas LI and LL. Figure 2b shows the same reversal for another session, while Fig. 2c shows a session where the probe spanned three areas (V1, LM, and LI), resulting in two reversals of retinotopy.

In each session, RF mapping was followed by presentation of nine 20-s-long movies, including six natural dynamic scenes, and three synthetic movies (a random sequence of white-noise patterns and Fourier phase-scrambled versions of two of the natural movies—see "Methods" and Supplementary videos). Four of the natural movies (termed "manual" below) were obtained by sweeping a hand-held camera at two speeds ("slow" and "fast") across an arena filled with 3D-printed geometrical shapes painted either black or white (example frames in Fig. 3a). The other two movies (termed "ratcam" below) were obtained by placing a camera on the head of a rat freely exploring an arena with the 3D-printed objects and another rat. Overall, 294 well-isolated units recorded in 18 rats were used in our analysis (see "Methods" for selection criteria): 168 from V1, 20 from LM, 36 from LI, and 70 from LL.

**Cortical representation timescales are stimulus- and area-dependent**. We first characterized the temporal structure of visual stimuli[61]. Briefly, for each movie, we computed the correlation coefficient between pixels belonging to image frames separated by a given lag. We fit the resulting dependence of the average correlation on the lag (Fig. 3b, solid line) with a decaying exponential (shaded areas). The exponential time constant defined the timescale of the movie (see "Methods"). This analysis showed that the dynamics of our stimuli spanned a range of timescales, from ~30 ms for the white-noise movie to ~600 ms for the slowest natural movie.

The activity in each neuronal population was stimulus-modulated, consistently across recorded neurons, for both fast and slow movies (Fig. 3c, where each line color-codes the response intensity of a LL (red) or V1 (green) unit to two movies). Thus, the overall population-averaged activity (green and red traces) was also strongly stimulus-modulated. To characterize how the temporal structure of the stimulus-locked responses depended on the visual input, we measured the population response timescale for each visual area and each movie by applying the same metrics defined above for the movie stimuli to the time-binned (bin width: 33 ms) and trial-averaged population response vectors[61]. Referring to Fig. 3c, we computed correlation coefficients between vertical slices (bins of spike counts) that were separated by a given lag (yellow frames), and then averaged the resulting coefficients to measure signal correlation[62] as a function of lag (Fig. 4a, solid lines; below we will term this "response correlation"). We fit the response correlation with an exponentially damped oscillation or a simple exponential based on a systematic model selection procedure (black dashed lines; see "Methods"), and the decay constant of the

exponential envelope was taken as the timescale of the response modulation. Next, we regressed the characteristic timescale of the response against the characteristic timescale of the stimulus, separately for each area, using a linear model with a common slope across all areas and an area-dependent intercept (Fig. 4b). The intercept is the baseline temporal timescale for stimulus-driven responses in each area. This baseline was clearly higher in the extrastriate areas (LM, LI, LL) than in V1, with smaller, statistically insignificant differences between LM, LI, and LL ($p = 0.4$ for both LM vs. LI, and LI vs. LL; two-tailed $t$-test, $t = 0.8$, $-0.8$, df = 31). Therefore, we repeated the regression analysis after aggregating these three areas (gray line). This revealed that response timescales in all areas depended strongly on the stimulus timescale (slope: $0.71 \pm 0.14$, significantly different from 0 at $p = 1e-5$, two-tailed $t$-test, $t = 5.2$, df = 33), and that the response timescale was significantly longer in extrastriate cortex than in V1 (intercept difference: $56 \pm 22$ ms; $p = 0.015$, two-tailed $t$-test, $t = 2.6$, df = 33). Overall, these results show that the characteristic temporal scale of the population representation of dynamic stimuli depends on the temporal scale of the visual input, and that representations in extrastriate cortex unfold over longer temporal scales than in V1.

**Intrinsic processing timescales increase along the ventral-like pathway**. As above, our results suggest that the timescale of stimulus-locked, trial-averaged neural representations increases from V1 to extrastriate cortex. The temporal scale of intrinsic neural processing may also increase across cortical hierarchies at the single cell[44] and population levels[45]. We tested whether this occurs in rat lateral visual areas by using the method described by ref. [44], which is mathematically similar to the procedure used above to compute the timescale of the population response vectors, but considers the responses of a single cell across multiple trials, rather than the average responses of multiple neurons ("Methods" and Supplementary Fig. 1). This allowed us to capture the largely stimulus-independent, within-trial temporal correlations in the spiking activity of a neuron, which were then averaged over all the units of a population. The time dependence of the resulting correlation function (solid lines in Fig. 4c) was fit with an exponential or an exponentially damped oscillation, based on systematic model selection performed independently for each condition (black dashed lines; see "Methods"). The temporal scale was characterized as the decay constant of the exponential envelope of the fit. Finally, we linearly regressed the intrinsic timescale of neural activity with the timescale of the movie stimulus and the cortical area (Fig. 4d), as we did above for the stimulus-driven response (Fig. 4b). The intercepts of the linear fit revealed a clear hierarchical organization of the timescale of intrinsic activity (V1: $36 \pm 14$ ms; LM: $120 \pm 14$ ms; LI: $203 \pm 14$ ms; LL: $213 \pm 14$ ms), with all extrastriate areas being significantly different from V1 ($p = 5e-7$, $1e-13$, $2e-14$, respectively, for LM, LI, LL, two-tailed $t$-test, $t = 6.3$, $12.6$, $13.3$, df = 31). We also found mild dependence on the timescale of the movie (slope: $0.17 \pm 0.07$, $p = 0.02$, two-tailed $t$-test, $t = 2.4$, df = 31), much weaker than observed for the stimulus-driven response (compare Fig. 3b, d). Interestingly, the range of intrinsic timescales recorded in our experiment quantitatively matched that reported for sensory cortex in behaving monkeys by ref. [44] and in behaving mice by ref. [45]. Overall, these results show that the temporal scale of the intrinsic activity increases along rat lateral extrastriate visual areas.

**Temporal representation stability grows over the ventral-like pathway**. Temporal stability of a neuronal representation supports stable perception of visual input that unfolds smoothly over

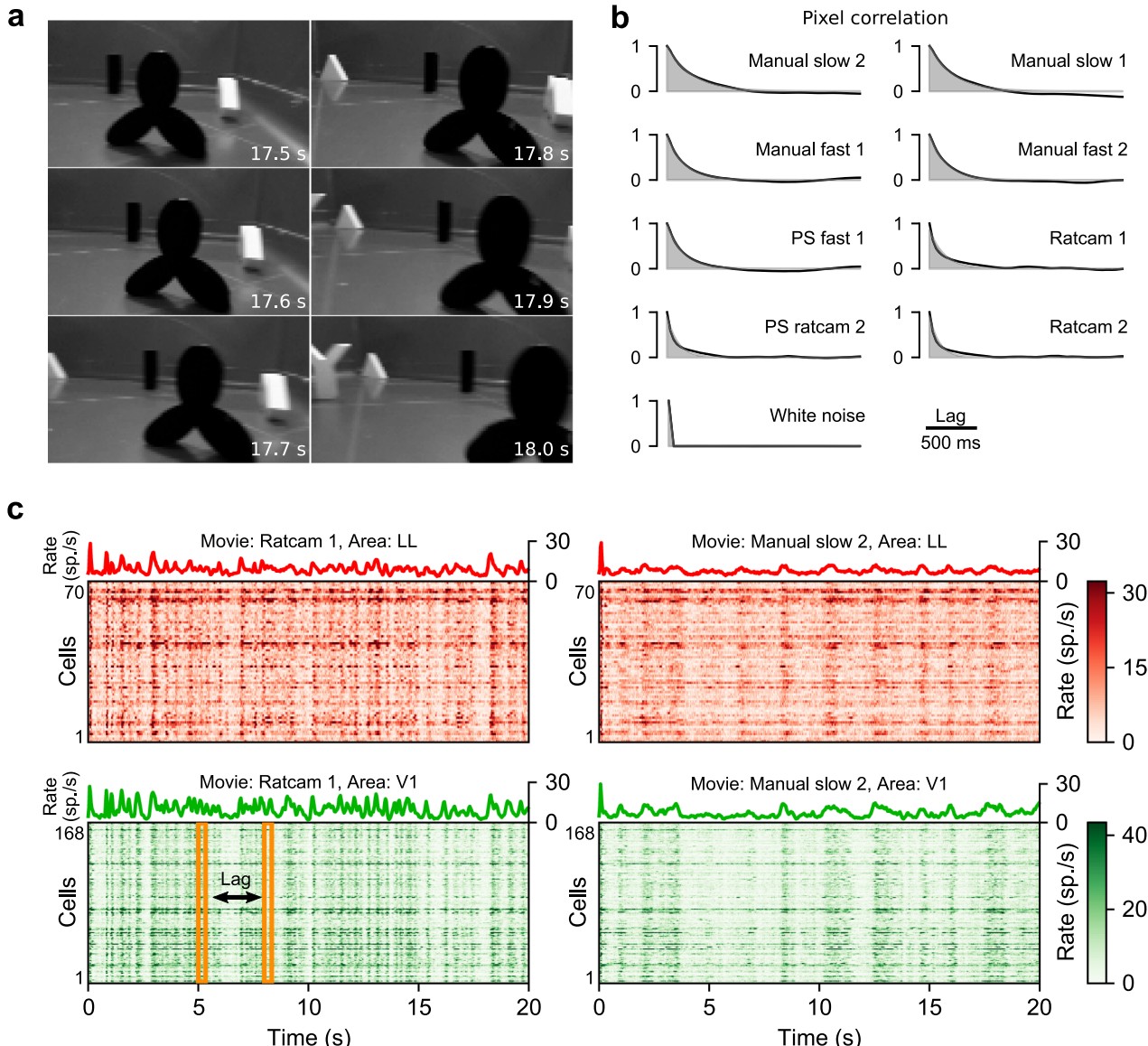

**Fig. 3 Visual stimuli and example neuronal recordings. a** Six consecutive example frames from one of the movies used as visual stimuli in our study. The numbers on the bottom right of each panel give the time at which each frame appears. **b** Quantification of the characteristic timescale of the individual visual stimuli (movies). The Manual and Ratcam labels refer, respectively, to movies that were taken either by moving a hand-held camera across an arena filled with 3D-printed objects (some of them visible in **a**), or by installing a small camera over the head of a rat that was allowed to roam inside an arena containing the 3D-printed objects and another rat. The PS label refers to Fourier phase-scrambled versions of the corresponding movies (see "Methods" for details). In each panel, the black line shows the average correlation of pairs of image frames within a movie as a function of their temporal lag, while the shaded area shows the best exponential fit. **c** Examples of population responses to two of the movie stimuli. In the color matrices, each row color-codes the intensity of the trial-averaged response (computed in 33-ms time bins) of a neuron belonging to either the LL (red) or V1 (green) population. The color-code is truncated to the 98th percentile of the firing rate distribution within each area for ease of display. The trace on top of each matrix shows the corresponding population-averaged response. The yellow frames illustrate the procedure to compute the characteristic timescale over which the population activity unfolded, in response to a given movie (see Fig. 4a, b). Source data are provided as Source Data files.

time. For instance, if an object (e.g., a triangle) sweeps across the visual field, a temporally stable representation will support discrimination of this object from other objects (e.g., squares) despite changes in its appearance. Correlation between neural activity at different times does not by itself assess such stability of discrimination, because population firing vectors can have a fixed amount of correlation and yet be discriminable to different degrees, depending on how they are organized with respect to a discrimination boundary. Moreover, a correlation measure assumes that deviations in the neural code are most important along directions that are orthogonal to the current population

vector, as correlation coefficients behave like a dot product or a cosine similarity measure. This is not necessarily the right notion of similarity for discrimination problems. Thus, we sought a direct test of the temporal stability of discrimination based on neural population responses.

To this end, we imagined a task where an observer learns to discriminate between visual input appearing at a time $t_A$ and that presented at a much later time $t_B$. If this task is performed on the basis of activity in a given cortical area, a trained classifier should be able to discriminate population response vectors at these two times. Now consider responses at a pair of time points shifted by

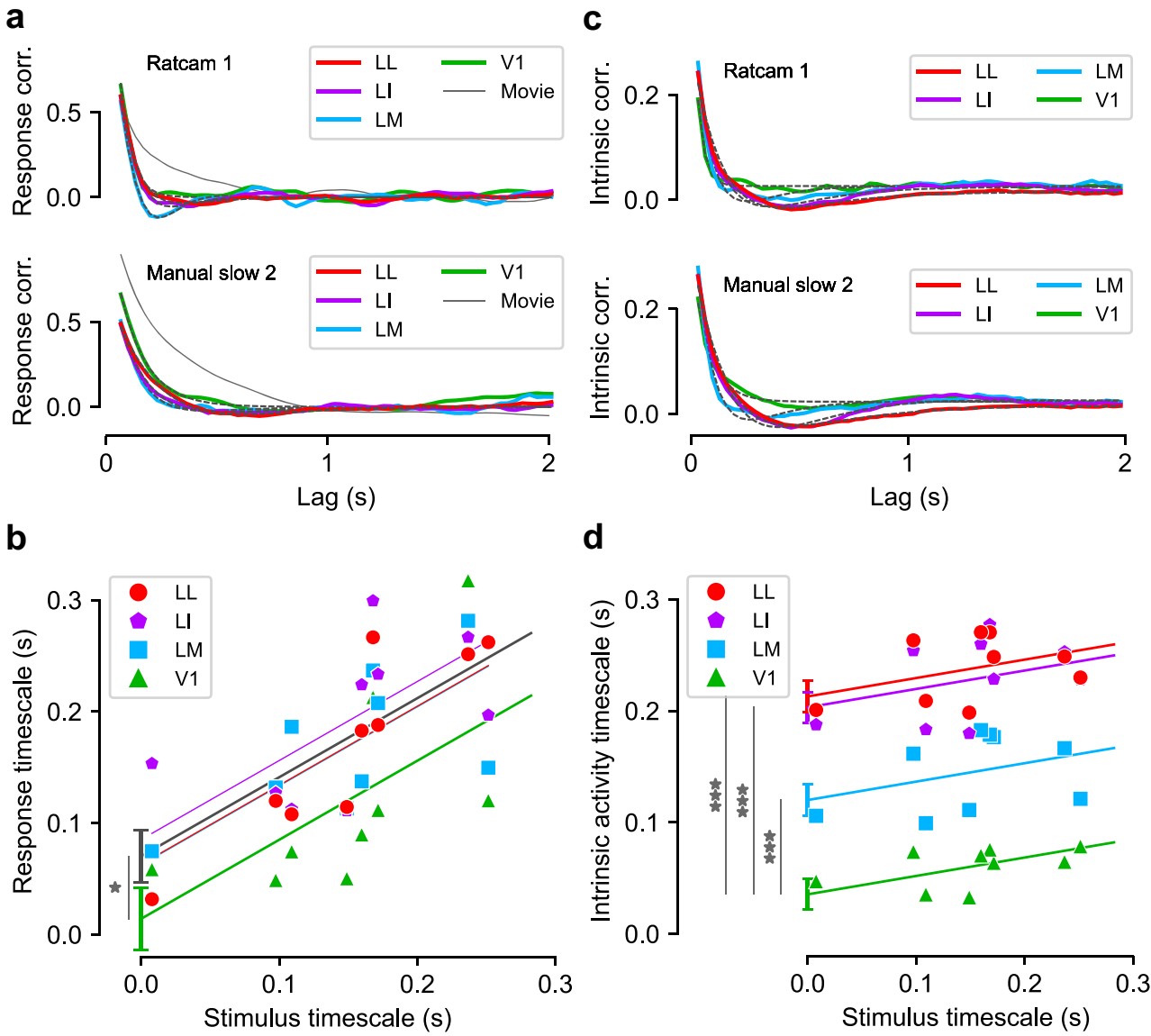

**Fig. 4 Characteristic timescale of the stimulus-driven population responses and of the intrinsic neuronal activity. a** Correlation between pairs of population response vectors (as those highlighted by the yellow frames in Fig. 3c) as a function of their temporal lag for two example movies (solid, colored lines). These curves were fitted with either an exponential decay or a damped oscillation function (fits are shown as black dashed lines). The time constants of these fits were taken as the characteristic timescales of the population responses. The solid gray lines show the correlation functions of the corresponding movies, reported from Fig. 3b for comparison. **b** Timescales of the population responses as a function of the timescales of the corresponding movie stimuli (colored markers). Each colored line is the linear fit prediction of the relationship between such timescales for a given area (same color-code as in the key; note that the lines for LM and LL are partially overlapping). The gray line is the linear fit prediction obtained by pooling together the data of the three extrastriate areas (i.e., LM, LI, and LL). Error bars: standard error of the intercept of the linear fits for V1 and the pooled extrastriate areas from regression analysis (see main text), $N = 36$ independent combinations of movies and areas. *$p = 0.015$, two-tailed $t$-test, no multiple test correction (see main text and "Methods" for details). **c, d** Same as **a** and **b**, but for the intrinsic timescales of neuronal activity (see main text for details). ***$p = 5e − 7$, $1e − 13$, $2e − 14$, respectively, for LM, LI, LL. Source data are provided as Source Data files.

a small lag $\Delta t$, namely $t_A + \Delta t$ and $t_B + \Delta t$. If the relevant part of the representation of the input is temporally stable over this lag, the trained classifier should perform as well on responses at the lagged time points as it did at the original times (schematic in Fig. 5a). Averaging over all $t_A$ and $t_B$ for each lag $\Delta t$ will assay how well the population responses support temporally stable discrimination of visual inputs.

Thus, we implemented a linear classifier on pseudopopulations of 20 randomly selected units (see "Methods"). Results for larger populations (50 units) from areas where these were available (V1 and LL) are reported in Supplementary Fig. 2. The classifier was

trained to distinguish population activities, represented as 20-dimensional spike-count vectors (bin width 33 ms), evoked by a movie at two different times. Critically, only a subset of trials was used for training (gray boxes in Fig. 5a). The temporal stability of the population code was then assessed by testing performance of the decoder at times shifted by a lag $\Delta t$ (Fig. 5a). To isolate contributions of the direct drive from the visual stimulus to overall temporal stability, we tested the classifier only on held-out experimental trials that were not used in training (green boxes). This ensured that intrinsic temporal correlations (Fig. 4d) between lagged frames would not affect similarity between

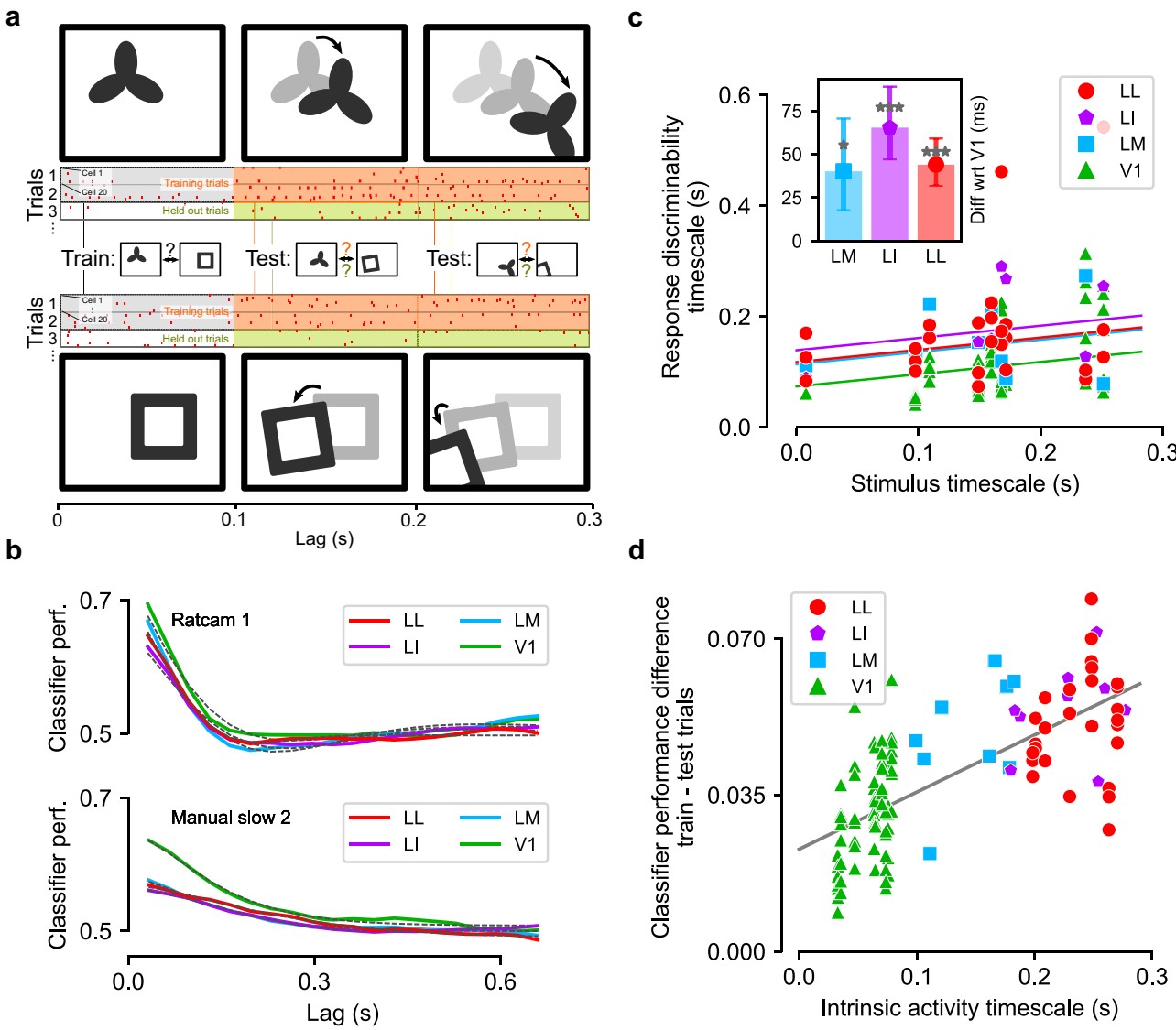

**Fig. 5 Temporal stability of neuronal representations, as assessed through decoding analysis. a** Schematic of the classifier analysis, aimed at assessing the temporal stability of the population representation. A linear classifier is trained to distinguish the population activity recorded at two different time points of the movie (spike counts computed from the gray time bins) and tested at other time points that follow (or precede, not shown) the training points by a certain lag. The classifier is tested on a held-out set of trials (green boxes), as well as on the same experimental trials that were used for training (orange boxes). The cartoons of the movie frames shown on top or below each time bin are for illustration purpose only: they highlight the fact that, in general, two nonoverlapping segments of a movie (from which the population activity is decoded) will contain different, transforming (e.g., translating, rotating, etc.) objects. For clarity, in this illustration, the size of the spike-count windows (i.e., gray, orange, and green boxes) was set to 100 ms, while 33-ms wide bins were used in the actual analysis. The spike patterns shown in the cartoon correspond to those actually fired by a randomly selected pseudopopulation of 20 units in V1 at two time points during the presentation of one of the movies (only three trials shown out of 30). **b** Classifier performance on held-out trials for two example movie stimuli and one example pseudopopulation per area (colored curves). The performance is plotted as a function of the lag between training bin (gray boxes in **a**) and test bin (green boxes in **a**), and is fitted with either an exponential decay or a damped oscillation function (fits are shown as black dashed lines). **c** Timescale of response discriminability, measured as the time constant of the exponential decay of the classifier performance on the held-out trial set, as a function of the timescale of the movies. Each dot corresponds to a distinct pseudopopulation of 20 units. The solid lines are linear regressions with common slope and different intercept across the four areas. The inset shows the difference of the intercept for areas LM, LI, and LL vs. area V1 (error bars are standard errors of the differences in the linear regression analysis). $N = 36$ independent combinations of movies and areas (*$p = 0.03$, ***$p < 1e-4$ for LI, $p = 2e-4$ for LL, one-tailed bootstrap test, $10^4$ bootstrap samples, no multiple test correction). **d** Amount of classifier performance due to intrinsic correlations ($\Delta_p$, Eq. (1)) as a function of the timescale of such correlations (i.e., the intrinsic timescale of neuronal activity shown in Fig. 4d). Source data are provided as Source Data files.

activity vectors used in training and testing. This analysis was repeated for each population and movie, and for all possible choices of pairs of training time points (see "Methods"). The results were averaged over these pairs to measure mean discrimination performance as a function of the lag (Fig. 5b; colored lines), and then fit with the same functional form as

above for the temporal correlations of neural responses (black dashed lines). Linear regression revealed that the decoder's performance depended significantly on the timescale of the movie stimulus (Fig. 5c; common slope: 0.22, 68% CI [0.15–0.32], nonzero at $p = 8e-4$, one-tailed bootstrap test). In addition, the intercept of the fit was significantly larger in the three extrastriate

areas than in V1 (V1 intercept: 74 [55-82]ms; difference with respect to V1 for the other areas: LM 40 [17-70] ms, nonnegative $p = 0.03$; LI 65 [48-89] ms, $p < 1e - 4$; LL 44 [32-59] ms, $p = 2e - 4$, one-tailed bootstrap test). This difference indicates that neural representations in higher areas of the ventral hierarchy support percepts of dynamic stimuli that are stable over longer timescales as compared to V1.

**Intrinsic correlations extend within-trial temporal representation stability.** Intrinsic correlations in neural firing fluctuations at different times quantify the tendency of neurons to sustain a given firing rate regardless of sensory input. As such, intrinsic correlations are argued to be a hallmark of longer stimulus integration times, or to act as "stabilizers" of neuronal representations to better support reading of the latter by downstream decision areas[44,45,63]. In vision, intrinsic temporal correlations (Fig. 4d) could stabilize the neural code against changes in visual input, maintaining a memory trace of objects moving across the visual field, thus stabilizing their perception.

To test this possibility, we repeated the analysis of the previous section. This time, however, we measured performance of the trained classifier at lagged time points in the same trials as used for training (orange boxes in Fig. 5a; performance curves in Supplementary Fig. 3a). In this case, fluctuations in test response vectors at times $t_A + \Delta t$ and $t_B + \Delta t$ will be correlated with fluctuations in the training vectors at $t_A$ and $t_B$. In fact, a neuron responding to a given movie frame in a given trial, with more or fewer spikes than its mean response across trials, will also tend to over- or under-fire in the following frames within the same trial. To quantify the contribution of such intrinsic correlations to discrimination stability, we focused on performance of the classifier at a lag of 33 ms (one video frame) from the training bin, and computed the performance gain from intrinsic correlations as follows:

$$\Delta_p = p_{\text{train}} - p_{\text{test}} \qquad (1)$$

Here $p_{\text{train}}$ and $p_{\text{test}}$ are performances of the classifier at lag 33 ms, when evaluated on the training trials (orange boxes in Fig. 5a) or on the held-out trials (green boxes), respectively.

As expected, given the progressively longer intrinsic correlations along the cortical hierarchy (Fig. 4d), $\Delta_p$ increased from V1 to LL (Fig. 5d. Slope of the fit: $0.13 \pm 0.01$, nonzero $p = 2e - 18$, two-tailed $t$-test, $t = 10.5$, df $= 115$; intercept: $0.023 \pm 0.002$, $p = 6e - 24$, two-tailed $t$-test, $t = 12.8$, df $= 115$), being, on average, almost twice as large in LL as in V1 (i.e., $0.051 \pm 0.002$ in LL vs. $0.03 \pm 0.001$ in V1). This revealed that intrinsic correlations play an increasing role in overall temporal stability of the neural code along the cortical progression. These findings show that the relative importance of intrinsic dynamics for population codes in visual cortex increases along the ventral-like, object-processing hierarchy.

**Temporal stability as a signature of invariance of neural representations.** The decoding analysis in Fig. 5 indicates that representations in rat lateral extrastriate areas remain stable over longer timescales than in V1. However, in terms of decoding performance, V1 surpassed all other areas, at least at time lags close to the training bin (Fig. 5b). In other words, the increase of invariance that is expected to take place along an object-processing hierarchy was observable only in relative terms (i.e., decoding performance changed less over time in LM, LI, and LL), rather than in absolute terms (since decoding performance was better in V1). This result can be understood on the basis of our previous study of rat lateral extrastriate cortex using static

objects[55], and by considering the movie segments that were used to carry out the decoding analysis in Fig. 5.

In ref. [55], we showed that LI and LL afford better decoding of object identity than V1 and LM under various transformations, but this effect only emerges when the objects to discriminate have similar luminosity across transformations. Otherwise, objects are actually better discriminated based on V1 than LL representations. This is because information conveyed by neural responses about low-level visual properties, such as luminosity and contrast, decreases substantially from V1 to LL. Such information pruning is consistent with ventral-like processing but has the unexpected effect of making it easier for V1 neurons to discriminate objects when their average luminance (across transformations) is different.

Given the stimulus set used in ref. [55] (bright, isolated shapes against a black background), it was possible to restrict decoding to object pairs with similar luminance, thus revealing the larger invariance of LL. This was not possible with our current, more naturalistic stimuli. Thus, differences in luminance and other low-level properties likely played a major role in allowing V1 to afford the best decoding performance. In fact, given the continuous nature of our dynamic stimuli, luminance and contrast differences between movie frames used to train a classifier would be preserved in the preceding and following frames. They would only vanish at long time lags from the training bins.

To verify correctness of this interpretation, we repeated our analysis with 21 pairs of movie segments of identical duration (967 ms, or 29 movies frames) that contained a single object (either black or white) moving roughly horizontally, either from left to right or from right to left (examples in Fig. 6a). This allowed defining four types of classification tasks, based on whether the color and/or motion direction of the objects to discriminate were (or not) the same (Fig. 6a, b). We expected that, at least for tasks where the difference in object luminance was small, extrastriate areas would reach the same absolute performance as V1 and possibly surpass it at sufficiently long lags from the training bin.

This new decoding analysis also allowed us to measure stability of visual representations in a scenario where object identity did not change within a movie segment. In fact, in our previous analysis (Fig. 5), segments fed to the classifiers were from all possible locations in the movie. Thus, every segment could contain multiple objects, and these could appear and disappear (being replaced by other objects) while the movie unfolded over time (e.g., Fig. 3a). While this analysis is ideal to obtain an overall, unbiased assessment of the stability of the cortical representation of a given movie, it does not allow measurement of temporal persistence of representations of individual objects undergoing identity-preserving transformations. It is the latter kind of stability that is expected to be equivalent to invariance measured with static objects and, as such, is expected to grow along an object-processing pathway. Therefore, the time constants of the decoding curves in Fig. 5 are likely an underestimate of the actual persistence (or invariance) of object representations in the four visual areas.

Our new analysis (Fig. 6b) largely confirmed these intuitions. In terms of decoding performance, V1 surpassed other areas at short time lags. However, the performance afforded by V1 neurons decayed abruptly over time, being matched and even surpassed by other areas at longer time lags. Notably, when objects had identical color but opposite direction (rightmost panel), the greater position invariance of extrastriate areas emerged clearly, with V1 performance substantially below that of the other areas at lags >300 ms. This confirmed that in tasks requiring invariant processing (e.g., for objects swapping position

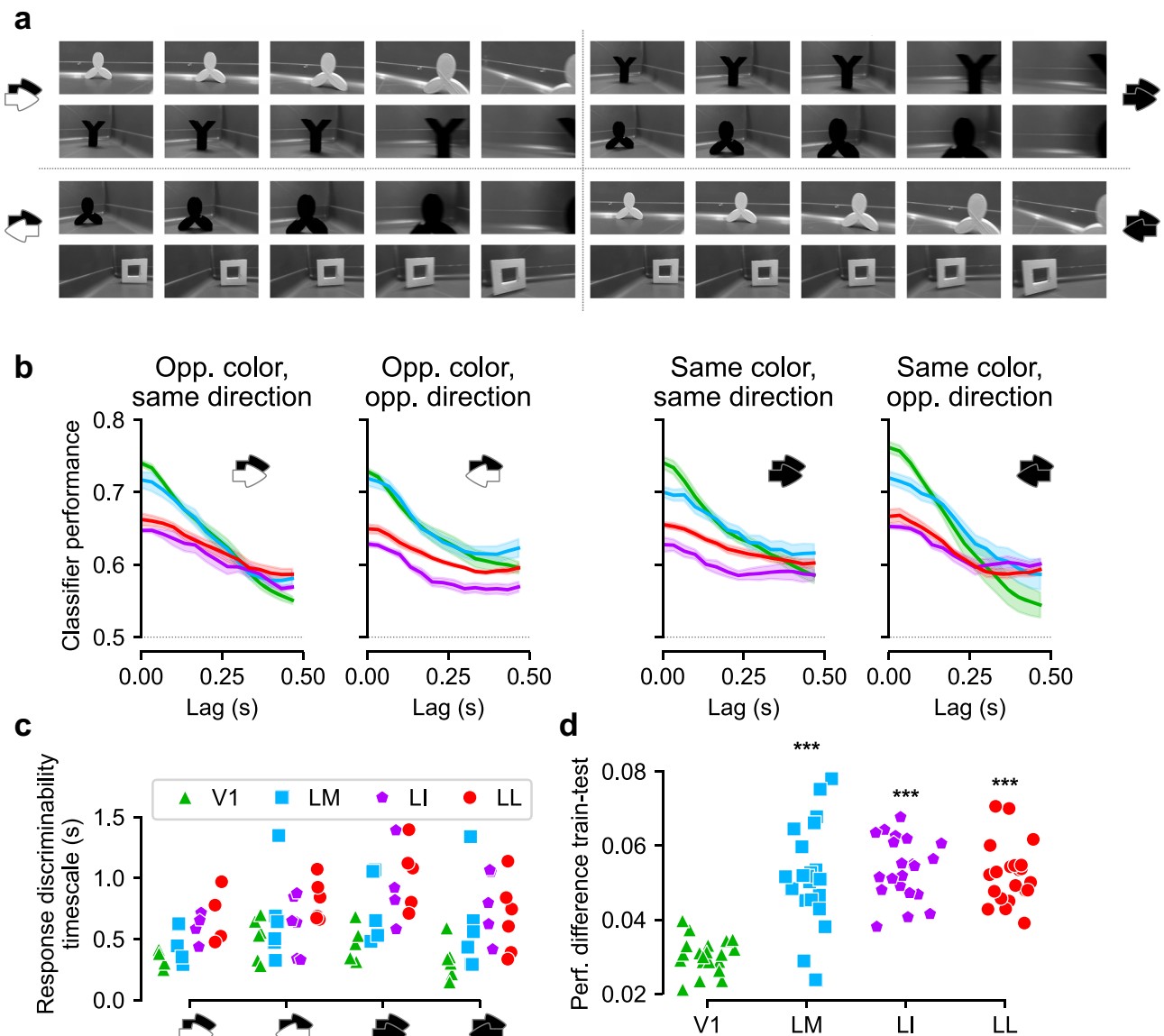

**Fig. 6 Classifier analysis restricted to hand-matched pairs of movie segments. a** Frames of four example pairs of movie segments (each pair is representative of one of the four classification conditions that were tested in our analysis). Each movie segment was exactly 29 frames long, and the frames shown here are frames number 1, 8, 15, 22, and 29 (left to right). As indicated by the black/white arrows in each quadrant, each pair of movie segments was distinguished from the others, based on whether the objects in the segments had the same color and whether they moved along the same direction. **b** Classifier performance vs. lag from the training bin for each of the four conditions. Solid line: average over all pairs tested in the task. Shaded area: s.e.m. **c** Timescales extracted from the decoding performances of all pairs of movies segments tested in any given classification task and for each area (two outlier data points are not shown: 3.65 s for same color/same direction, and 2.73 s for same color/opposite direction). **d** Amount of classifier performance due to intrinsic correlations (difference between performance on training trials and on test trials; as defined in Fig. 5d) for each movie pair and each area. $*p = 1e − 7$, $7e − 10$, $3e − 9$ for LM, LI, LL, respectively, paired $t$-test on each area vs. V1. Source data are provided as Source Data files.

in the visual field) and less prone to solution based on luminance differences (e.g., objects with identical color), lateral extrastriate areas emerge as most invariant[55].

Restricting the decoding analysis to segments with single objects resulted in a much slower dynamics (compare performance curves in Figs. 5b and 6b). This increased temporal stability of object representations was pronounced for high-order extrastriate areas (LI and LL), and less so for LM. In V1, instead, the representation unfolded much faster. Fitting exponentially damped oscillations to performance curves yielded time constants (Fig. 6c) that, in LI and LL, were up to five times larger than obtained in our previous, unconstrained analysis (Fig. 5c). Importantly, time constants in LI and LL were ~2× as large as

in V1, revealing substantially slower unfolding of higher-order representations. A statistical test based on categorical linear regression (see "Methods") confirmed that time constants in LI and LL were higher than in V1 (difference with respect to V1: LM, 210 ± 140 ms; LI, 550 ± 140 ms; LL, 390 ± 140 ms; $p$ values, two-tailed $t$-test: LM 0.13, LI 0.0001, LL 0.006).

In Fig. 6b, c, as previously done in Fig. 5b, c, the generalization ability of classifiers was tested on held-out trials not used for training. To gauge the impact of intrinsic temporal correlations in stabilizing object representations, we measured generalization at lagged time points in the same trials as used for training (same analysis as in Fig. 5d). As before, the performance increment $\Delta_p$ (Eq. (1)) afforded by intrinsic correlations was substantially and

significantly higher in lateral extrastriate areas than in V1 (Fig. 6d; mean $\Delta_p$: V1, $0.030 \pm 0.001$; LM, $0.052 \pm 0.003$; LI, $0.054 \pm 0.002$; LL, $0.052 \pm 0.002$; paired $t$-test $p$ values for the difference with respect to V1: LM $p = 1e - 7$, LI $7e - 10$, LL $3e - 9$). This confirmed the increasingly important role played by intrinsic correlations in determining the overall temporal stability of object representations along the cortical progression.

**Temporal stability of visual cortical representations in awake mice.** The results in previous sections are based on recordings in anesthetized rats. To validate in awake animals, we analyzed a large Allen Institute dataset of neural responses recorded from many visual areas of awake mice using Neuropixels probes[12]. These recordings also provided the opportunity to test our conclusions in a different rodent species. Furthermore, the ranking of mouse visual areas in terms of hierarchical processing has been carefully established anatomically[64,65] and quantified by the definition of an anatomical hierarchy score[66]. This score is related to several functional measures of hierarchical processing, thus checking consistency between anatomy and physiology[12].

Thus, we sought to establish whether, in the Allen dataset, temporal stability of neural responses to dynamic stimuli increased as a function of the hierarchy score of mouse visual areas. To this end, we focused on two natural movies (one lasting 30 s, presented 20 times; the other lasting 120 s, presented 10 times) that were part of the visual stimulus battery tested in ref. [12]. The shorter clip was discarded, given that its pixel correlations displayed a strongly irregular pattern (see "Methods" and Supplementary Fig. 4a, left), likely because the clip is a sequence of highly static scenes. We thus carried out our analyses on the longer clip, whose pixel correlation function has an exponential decay (Supplementary Fig. 4b, green curves) that is fully comparable with our stimuli (see Fig. 3b). We extracted the neuronal responses to repeated presentations (trials) of this movie recorded in the first 17 sessions of the Allen dataset from the lateral geniculate nucleus (LGN) in the thalamus (571 cells), V1 (876 cells), and five higher-order, extrastriate visual cortical areas: LM, AL, RL, PM, and AM (506, 511, 796, 565, and 687 cells respectively). For each area, we computed the correlation function of the population response vectors (same analysis as Fig. 3c), as well as the population-averaged intrinsic correlation function, capturing within-trial fluctuations of the firing rate (same analysis as Supplementary Fig. 1).

The resulting correlation functions are shown in Fig. 7a, b (solid lines), with best exponential fits to the data (dashed lines; same procedure as Fig. 4a, c; see "Methods"). The fits gave time constants of the exponential decays across the seven areas and we regressed these against the corresponding hierarchy scores. For both stimulus-driven (Fig. 7c) and intrinsic (Fig. 7d) time constants, we found a clear, significant linear relationship (respectively, $p = 0.001$, $t = 5.8$, and $p = 0.01$, $t = 3.3$; one-tailed $t$-test, df $= 5$). That is, we observed increased temporal stability across the visual hierarchy of awake mice that was very similar to our finding across the object-processing pathway of anesthetized rats.

On the whole, the time constants measured for the Allen dataset were substantially larger than those in our experiments. This is unsurprising, given that the Allen movie clip unfolded over time with much slower dynamics than the movies used in our recordings (compare Supplementary Fig. 4 to Fig. 3b) and, based on trends in Fig. 4b, d, we expected the dynamics of the response to track that of the stimulus. As shown in Fig. 7e, f, the latter conclusion was confirmed when the time constants obtained in our recordings (green dots; same data of Fig. 4b, d) and those extracted from the Allen dataset (red dots; same data of

Fig. 7c, d) were plotted against time constants of the corresponding movies. The dependence between response and stimulus dynamics inferred from our recordings extrapolated well to the Allen dataset, with the red dots being very close to values expected based on a linear regression (dashed green line) performed on the green dots only. This is striking, considering the narrow span of temporal scales in our movies (0–0.25 s), compared to the time constant of the Allen clip (2.8 s).

**Dependence of temporal stability on the behavioral state of the animal.** The results above indicate that our conclusions are robust across rodent species (i.e., rat vs. mouse), the processing hierarchy under exam, and the state of the animal (i.e., anesthetized or awake). However, awake states can be further separated into a spectrum of finer states, depending on alertness, motivation, appetite, or activity of the animal[46,48–51]. In particular, recent studies have shown that locomotion increases response magnitude and spatial integration in mouse visual cortex[46,48]. Importantly, during active wakefulness, mouse visual cortical representations of natural movies have been found to be more similar to those measured under anesthesia than during quiet wakefulness[49].

Motivated by this, we noted that mice in the recordings were head-fixed but were free to either rest or run on a spinning wheel, whose angular velocity was part of the data. We used this to identify epochs of neural response during which the mouse was either resting or running (see "Methods"). We then computed response and intrinsic timescales separately for resting and running states. The trends changed slightly depending on the number of trials $N_t$ (i.e., stimulus repetitions) in the analysis and the minimal duration $L$ of resting (or running) epochs shared among the chosen trials. Given that these two parameters were inversely related ("Methods" and Supplementary Fig. 5), we considered all combinations of four choices of $N_t$ (i.e., 3, 4, 5, and 6 trials) and five choices of $L$ (i.e., 2.5, 3.3, 4.1, 5.0, and 6.6 s).

Overall, we found that the temporal stability increase observed across the mouse visual hierarchy was weaker in the resting than the running state. Figure 8 (leftmost plots) shows an example for one choice of $N_t$ and $L$: the slope of the linear fit through time constants is smaller in the resting state, especially for intrinsic dynamics (Fig. 8b). Figure 8a (right) shows that for response timescales, the reduction of the fit slope from the resting to the running state is observed across most of the $4 \times 5$ combinations of the parameters $N_t$ and $L$, yielding a significant difference (two-tailed, paired $t$-test; $p = 0.002$, $t = 3.6$, df $= 19$). The effect is even stronger for intrinsic timescales (Fig. 8b, right), where the slope of the fit decreases by more than half in the resting as compared to the running state across all combinations of $N_t$ and $L$, yielding a highly significant difference (two-tailed, paired $t$-test; $p = 2e - 8$, $t = 9.2$, df $= 19$).

Overall, this analysis shows that, during quiet wakefulnes, temporal stability increase across the visual processing hierarchy is substantially reduced as compared to the state of active wakefulness, especially for intrinsic correlations. To further study this phenomenon, we analyzed a third dataset of neural responses to natural movies collected in head-fixed and body-restrained awake rats in a previous study of the rat object-processing pathway[60]. In this study, neural responses were obtained from two of the regions also sampled in our recordings, i.e., V1 (44 cells) and LI (40 cells), plus a third visual area, named TO (38 cells), that follows LL in the anatomical progression of lateral extrastriate areas and that a previous study indicates as a part of the rat ventral-like pathway[56].

In this head-fixed rat dataset, autocorrelations of the stimulus-driven responses displayed exponential decays with time

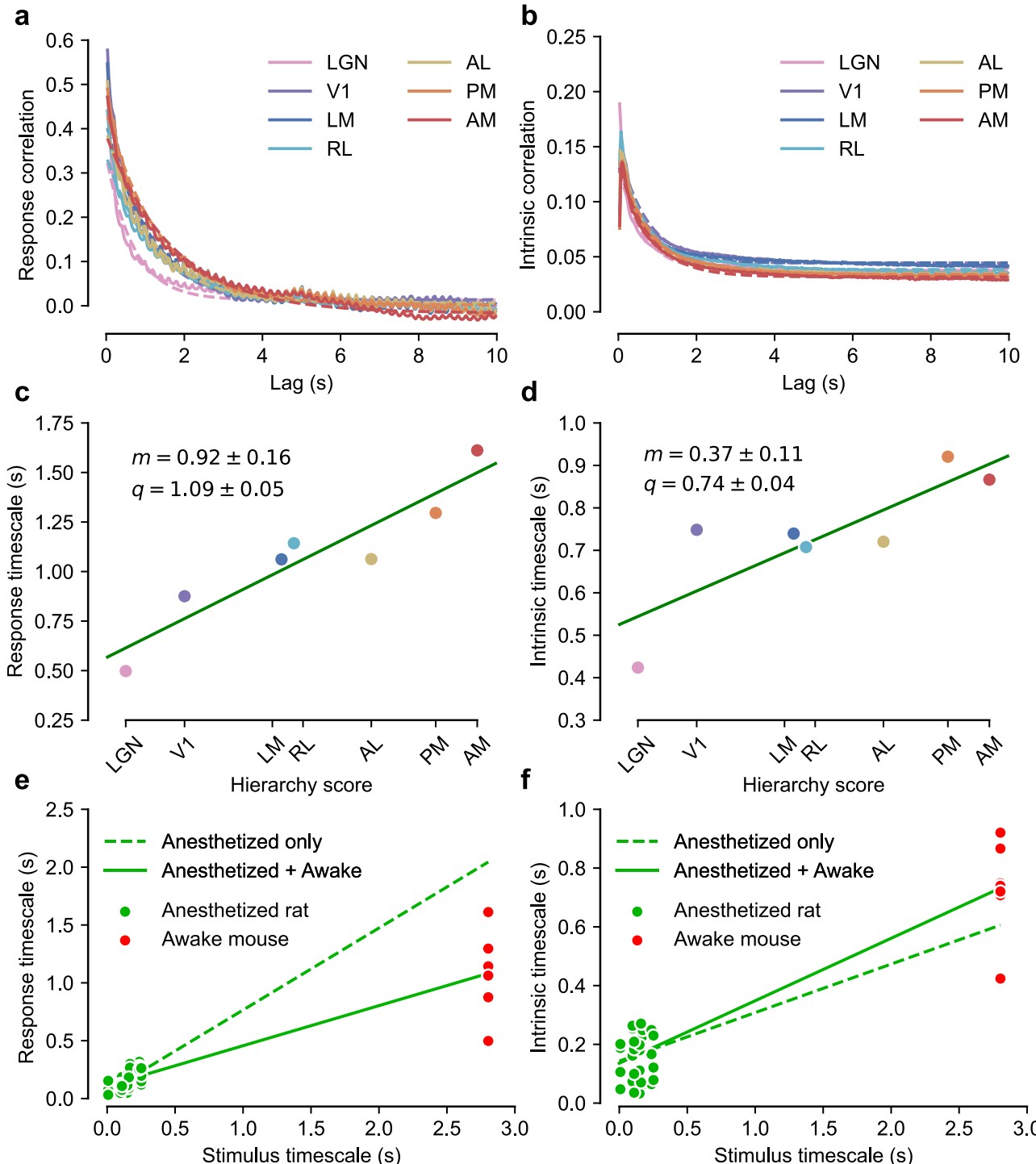

**Fig. 7 Timescale of stimulus-driven responses and intrinsic neuronal activity in awake mice (Allen dataset). a** Correlations in the responses to a 120-s-long movie clip that were recorded across seven visual areas of the mouse visual system by ref. [12] and are part of the Allen dataset. The same color-code used by ref. [12] denotes the different areas (see key). Dashed lines with matching colors indicate the corresponding exponential fits. **b** Same as in **a** but for the intrinsic correlations. **c** The time constants of the exponential fits to the correlation functions shown in (**a**) are plotted against the anatomical hierarchy score of the corresponding visual areas, as measured by ref. [68]. *m*: slope of the best linear fit. *q*: intercept of the best linear fit. **d** Same as in **c** but for the intrinsic correlations shown in **b**. **e** All the response timescales (without area distinction) obtained in anesthetized rats (green dots; same data as in Fig. 4b) and in awake mice (red dots; same data as in **c**) are plotted against the corresponding stimulus timescales. The dashed green line shows the linear regression performed just on the rat anesthetized data, while the solid line shows the regression performed on the combined data. **f** Same as in **e** but for the intrinsic timescales previously shown in Fig. 4d for the anesthetized rat recordings (green dots) and in **d** for the awake mouse recordings (red dots). Source data are provided as Source Data files.

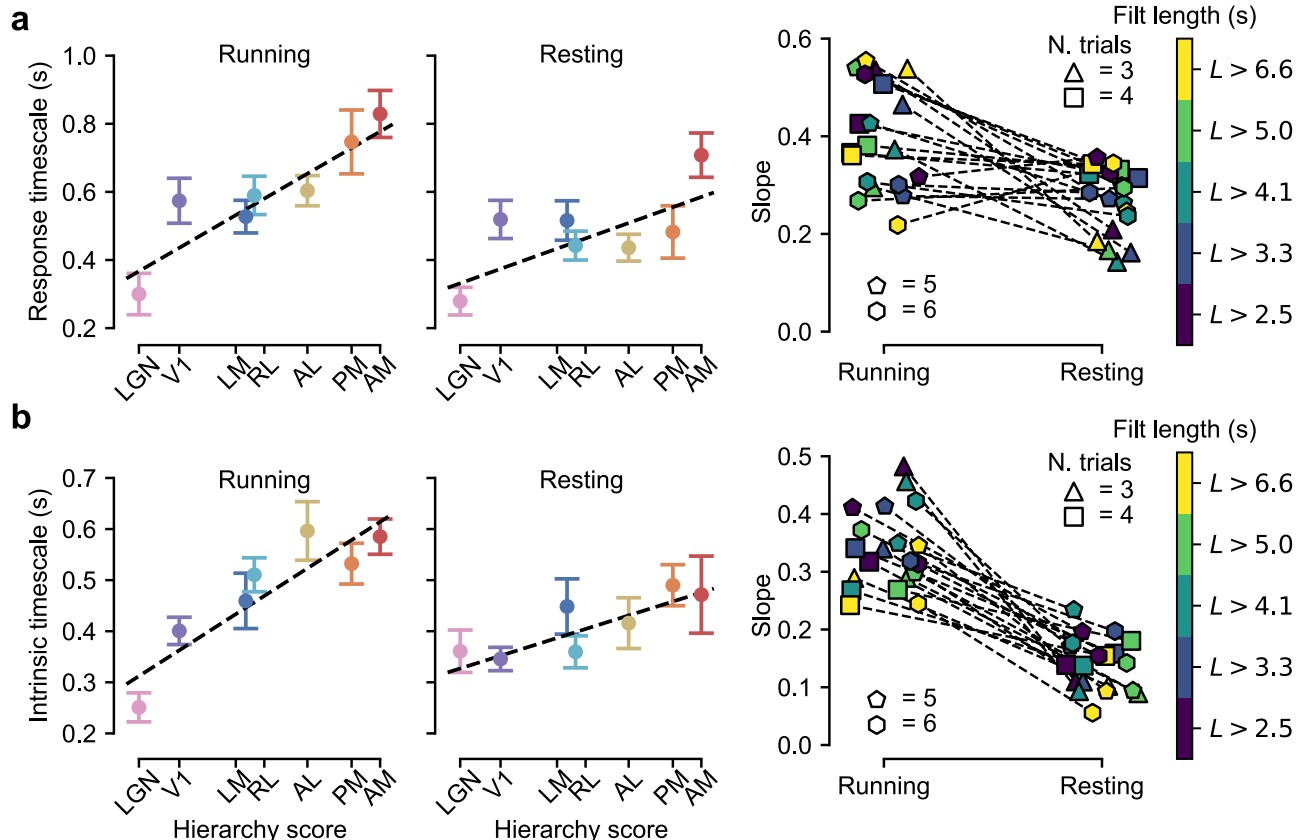

**Fig. 8 State dependence of response and intrinsic timescales. a** Left, center: examples of the hierarchical trends of response timescales as a function of the anatomical hierarchy score for the particular choice of the parameters $N_t = 5$ and $L = 4.125$ s (see main text) in the running (left) and resting (center) state. Data points are averages over 11 epochs of running and resting states. Error bars are s.e.m. The dashed black lines are error-weighted linear regressions. Right: scatter plot reporting the angular coefficients of the linear regressions for all the $4 \times 5$ combinations of the parameters $N_t$ and $L$, in the case of the running and resting states (the dashed lines connect matching choices of $N_t$ and $L$). The shape of the markers codes for $N_t$, the color for $L$ (see key). **b** Left, center: same as in **a** (left, center) but for the intrinsic timescales and for the choice of the parameters $N_t = 6$ and $L = 2.475$ s. Data points are averages over 11 epochs of running and resting states. Right: same as in **a** (right) but for the intrinsic timescales. Source data are provided as Source Data files.

constants similar in high-order areas (i.e., LI and TO) and in V1 (Supplementary Fig. 6a, b). In addition, the dependence between the temporal scales of the response and of the movie was weak and not significant (regression slope: $0.07 \pm 0.07$, $p = 0.3$, $t = 1.04$; intercept difference extrastriate-striate: $-0.015 \pm 0.020$, $p = 0.46$, $t = -0.74$; df = 57; see "Methods"). The intrinsic correlations were also weaker than in our recordings and in the Allen data, and did not display a monotonic decrease with time (compare Supplementary Fig. 6c to Figs. 4c and 7b). We nevertheless fit exponential decays to the tails of these autocorrelation functions. The resulting time constants did not increase from V1 to higher-order areas (Supplementary Fig. 6d; regression slope: $0.0 \pm 0.2$, $p = 0.99$, $t = -0.02$; intercept difference with respect to V1: LI $-0.02 \pm 0.06$, $p = 0.7$, $t = -0.36$; TO $-0.01 \pm 0.06$, $p = 0.89$, $t = -0.15$; df = 56), thus behaving differently from our recordings (Fig. 4d) and the Allen data (Figs. 7d and 8b, left).

Given that the movies in ref. [60] were much shorter (5 s) than those in our experiments (20 s) and in the Allen study (120 s), and fewer trials were collected (10) as compared to our recordings (30), we checked whether these results could be explained by a lack of statistical power in the head-fixed rat dataset to detect trends in our recordings and the Allen data. Thus, we replicated our analysis in Fig. 4b, d, after subsampling the number of available trials and progressively reducing the duration of the responses included in the analysis (see Supplementary Information and Supplementary Fig. 7). We found that shortening the

movie (and, to a lesser extent, the trial number) strongly reduced sensitivity to dependence of the response timescale on that of the movie, as well as to increase of response stability across the visual hierarchy (see Supplementary Fig. 7a, b). Conversely, sensitivity to differences between intrinsic temporal scales of extrastriate lateral areas and those of V1 was only marginally reduced (see Supplementary Fig. 7d–f). This suggests that the lack of increase in the intrinsic timescales from V1 to LI/TO in the head-fixed rat dataset is a genuine difference with our recordings in anesthetized rats and with the Allen experiments in awake, behaving mice—a difference that is fully consistent with the reduced increase of intrinsic temporal scales observed in the Allen data when mice rested rather than running (Fig. 8b).

**Discussion**
We tested the stability of neural representations along the rodent analog of the ventral stream in anesthetized rats viewing naturalistic and synthetic movies. We asked whether the temporal stability of visual cortical representations increased along the hierarchy under dynamic stimulation, as suggested by the build up of invariance observed using static stimuli[1,5,9,21]. Any effect of this kind could be suppressed by adaptive and top–down mechanisms (e.g., prediction error signals) in visual cortex[26,67] that favor encoding of surprising or transient inputs over predictable or sustained ones[12,30,33,40,68]. The potential role of such mechanisms in representing dynamic inputs can be studied by

simulating a recent model of neural adaptation[42], which we extended by adding neuronal noise and intrinsic integration (see Supplementary Information). The model suggested that adaptive mechanisms could in principle suppress increases in timescale expected for more persistent stimuli, thus possibly equalizing response dynamics along the visual cortical hierarchy (Supplementary Fig. 8e). However, we found that, in all tested visual areas, the temporal scale of visual encoding depended not only on characteristic stimulus timescales, but also increased significantly from V1 to extrastriate visual cortex (Figs. 4a, b and 5b, c). This increase was particularly strong when assayed in terms of the ability of cortical representations to sustain discrimination of movie segments of similar duration containing single objects translating along either similar or opposite trajectories within the visual field (Fig. 6b, c). This implies that the increase of invariance afforded by feedforward computations along the rat ventral pathway[1] is not washed out by possible adaptive, recurrent and top–down processes (see "Introduction"). Thus, when the pathway is probed with dynamic stimuli, a functional hierarchy of representational timescales is observed that parallels the hierarchical buildup of invariance revealed by brief presentation of static stimuli[55,56].

An alternative explanation might be that this trend exclusively reflects a systematic increase of the width of the filters describing the temporal integration of low-level visual features by neurons along the hierarchy. In the Allen dataset[12], the authors estimated spike autocorrelation functions of recorded neurons during responses to flash stimuli, from which they extracted a characteristic time constant by fitting a decaying exponential. They found that the resulting time constants increase along the visual hierarchy. Could this increase explain the growth of the response timescale to movies that we analyze, given that the two temporal scales are significantly correlated (see Supplementary Fig. 9a; $r = 0.75$; $p < 0.01$, one-tailed $t$-test), and even though they do not necessarily measure the same quantity? To test this, we modeled each recorded neuron in the Allen dataset as a linear filter, with spatial kernel given by its RF profile and increasingly broad exponential temporal kernels defined by time constants estimated in ref. [12]. By convolving the input signal (i.e., the 120-s movie) with these spatiotemporal linear filters, we obtained a crude simulation of responses of the units to the movie. The resulting response timescales, estimated as in our analyses, did not differ much between areas. Thus, we did not observe any monotonic increase as a function of the anatomical hierarchy score (Supplementary Fig. 9b, c). This suggests that the temporal filtering properties of rodent visual neurons are not sufficient to explain the observed hierarchical increase of response timescales under dynamic stimulation. This conclusion is supported by the observation that cortical representations were up to five times more stable when rat high-level neurons (in LI and LL) were tested with movie segments containing single objects (Fig. 6b, c) than random movie segments (Fig. 5b, c). Given that the neurons compared in these analyses were the same, they necessarily had identical time filtering properties. This suggests that the fivefold increase of stability is a result of nonlinear processes, such as those underlying invariance, that maximally display their power to stabilize cortical representations when neurons are probed with single objects undergoing identity-preserving transformations (Fig. 6).

Our study also addresses the role of activity-dependent processes that can extend persistence of neural representations along an object-processing pathway, affecting temporal spans over which firing rate fluctuations are correlated within single trials. In both primates and rodents, temporal scales of these intrinsic fluctuations increase along various cortical hierarchies[11,43–45]. Mechanistically, the nature of these intrinsic processes remains unclear, but most authors attribute them to temporally extended input integration or recurrent computations, meant to sustain the neural representation over timescales that guide perception and behavior[44,45,63]. Experimentally, however, most work in the ventral stream has focused on static stimuli, and thus it is not known whether intrinsic dynamics contribute to the structure of population codes for stimuli that are themselves varying in time.

Our experiments found strong evidence that intrinsic timescales increase along rat lateral extrastriate areas (Fig. 4c, d), and that this increase helps to stabilize neural representations of visual inputs along the cortical hierarchy (Figs. 5d and 6d). We also found weak but significant dependence of the intrinsic timescales on stimulus dynamics (Fig. 4d)—consistently with the magnitude of intrinsic timescales measured in awake mice (Fig. 7f). This means that intrinsic fluctuations of neural firing are correlated over a time span related to the overall rate of change of the visual input. Thus, although our experiments did not investigate mechanisms behind such intrinsic, activity-dependent processes, they clarify their possible functional role and relationship with the overall dynamics of the visual input.

To achieve control and repeatability of stimuli within each neuron's RF over extended periods (1.5/2 h), we performed acute recordings under anesthesia. We expected, in view of previous work in anesthetized animals[49,61], that spatiotemporal characteristics of population activity that we measured would be similar in awake animals. To verify, we repeated our assessment of the temporal stability of neural firing during presentation of natural movies using the Allen dataset[12]. These data allowed us to test a different species (mouse), the awake state, and another anatomically well-established visual hierarchy. This hierarchy plays a broader functional role than the rat lateral progression, encompassing both object- and motion-processing areas[53,54,58,65,66]. In addition, by partitioning responses into resting and running epochs, we compared dynamics of neural firing during active and quiet wakefulness. We found that the increase of timescales for stimulus-driven responses and intrinsic fluctuations is reproduced along the mouse visual hierarchy (Fig. 7c, d). The hierarchical increase of response timescales was actually more evident than in rat lateral extrastriate areas. This is likely because the Allen dataset also includes LGN (a subcortical region) and encompasses a larger number of visual cortical areas, thus allowing tests of the temporal response scale across a deeper processing hierarchy. Interestingly, the fact that this hierarchy underlies both ventral- and dorsal-like functions suggests that increasing the temporal stability of visual representations is an overarching goal of the whole visual system. This is consistent with the fact that hierarchical growth of invariance is necessary to support perceptual constancy not only of object identity but, generally, of any visual attribute, such as color or motion direction and speed[69].

We found that the increase of temporal scales was attenuated (especially for intrinsic dynamics) during quiet wakefulness, as compared to active wakefulness (Fig. 8). To investigate the state dependence of our findings, we analyzed a third dataset of recordings in awake, body-restrained rats[60], and found no evidence of an increase of response and intrinsic timescales from V1 to extrastriate lateral areas (Supplementary Fig. 6). The failure to detect a growth of the response timescales could be explained in part by low statistical power (Supplementary Fig. 7), but we showed that this was not the case for intrinsic timescales. Therefore, the fact that they remained stable across the lateral processing hierarchy appears to be a result of the enforced immobilized state—a finding that is compatible with the attenuation of the hierarchical growth of intrinsic timescales observed in the Allen dataset during resting bouts (Fig. 8b).

Our results fit previous studies reporting weakened processing of the visual input by rodent visual neurons (e.g., in terms of

response magnitude, spatial integration, sparseness, reproducibility, and discriminatory power) during quiet wakefulness, as compared to active wakefulness and the anesthetized state[46,48,49]. The shortened temporal processing scales observed in our study adds to this list of attenuated visual processes under quiet wakefulness. Future studies at the circuit/synaptic level should aim to clarify whether these phenomena have a common mechanistic origin or whether one causally implies the other. At the functional level, these observations have led some authors to conclude that during quiet wakefulness visual cortex may be perceptually disengaged or detached from the visual environment, even relative to the anesthetized state[46,49]. The reduced (Fig. 8b) or canceled (Supplementary Fig. 6c, d) hierarchical growth of the intrinsic timescales observed in our analyses during, respectively, voluntary or enforced stillness, is consistent with this interpretation. In fact, during quiet wakefulness, there is arguably no need to regulate the temporal span over which the incoming visual signal is integrated to guide perceptual decisions and motor behavior, especially during enforced body restraint. As such, it may be computationally and metabolically efficient for the brain to turn off those activity-dependent, intrinsic processes that set the proper timescales of signal integration across the various stages of the visual processing hierarchy.

What roles could these intrinsic dynamics play in behavior? Intrinsic processing may be crucial for perception in a noisy, changing environment. For example, maintenance of sensory information by stimulus-independent temporal correlations in population activity can lead to better behavioral performance, when consistent estimates of a quantity are needed[45,63]. Alternatively, intrinsic processing may support predictive coding[39,41], allowing neural circuits to use feedforward inputs to predict and represent what will happen next, an ability with obvious utility for behavior. In order to play these roles effectively, the dynamics of intrinsic processing should be adapted to the temporal structures that are encountered in natural environments. Previous work has shown (or at least suggested) that many aspects of spatial and temporal processing in the early visual system are indeed adapted to the structure of natural scenes. Examples are histogram equalization[70] and on–off circuit asymmetries in the retina[71], development of V1 simple and complex cells[59,72–75], texture perception[76], and eye movements[77]. The idea that intrinsic processing could be similarly adapted deep into the cortical hierarchy could be causally tested by altering the animals' visual environment during development or while learning a task, and then measuring neural population activity across the cortical hierarchy with ethologically relevant, dynamic stimuli.

## Methods

**Animal preparation and surgery**. All animal procedures were in agreement with international and institutional standards for the care and use of animals in research and were approved by the Italian Ministry of Health: project N. DGSAF 22791-A, submitted on September 7, 2015 and approved on December 10, 2015 (approval N. 1254/ 2015-PR). Eighteen male Long Evans rats (Charles River Laboratories) with age 3–12 months and weight 300–700 g were anesthetized with an intraperitoneal (IP) injection of a solution of 0.3 mg/kg of fentanyl (Fentanest®, Pfizer) and 0.3 mg/kg of medetomidine (Domitor®, Orion Pharma). Body temperature was kept constant at ~37° with a warming pad, and a constant flow of oxygen was delivered to the rat to prevent hypoxia. The level of anesthesia was monitored by checking the absence of tail, ear, and hind paw reflex, as well as monitoring blood oxygenation, heart, and respiratory rate through a pulse oximeter (Pulsesense-VET, Nonin), whose sensor was attached to one of the hind paws. After induction, the rat was placed in a stereotaxic apparatus (Narishige, SR-5R) in flat-skull orientation (i.e., with the surface of the skull parallel to the base of the stereotax), and, following a scalp incision over the left and posterior parietal bones, a craniotomy was performed over the target area in the left hemisphere (typically, a 2 × 2-mm² window). Dura was also removed to ease the insertion of the electrode array. Stereotaxic coordinates for V1 recordings ranged from −7.49 to −8.36-mm anteroposterior (AP), with reference to bregma; for extrastriate areas (LM, LI, and LL), they ranged from −6.64 to −7.74-mm AP.

Once the surgical procedure was completed, the stereotaxic apparatus was moved on top of an elevated rotating platform. The right eye was immobilized with an eye ring anchored to the stereotaxic apparatus, and the left one was covered with opaque tape. The platform was then rotated, so as to align the right eye with the center of the stimulus display and bring the binocular portion of its visual field to cover the left side of the display. Throughout the experiment, ophthalmic solution Epigel (Ceva Vetem) was regularly applied onto the right eye and the exposed brain surface was covered with saline to prevent drying. In addition, a constant level of anesthesia was maintained through continuous IP infusion of the same anesthetic solution used for induction, but at a lower concentration (0.1-mg/kg/h fentanyl and 0.1-g/kg/h medetomidine), by means of a syringe pump (NE-500; New Era Pump Systems).

**Neuronal recordings**. Extracellular recordings were performed with 32-channel silicon probes (Neuronexus Technologies), following the same procedure used in ref. [55]. Briefly, to maximize the RF coverage, recordings in V1 were performed with 4- or 8-shanks probes, which were inserted perpendicularly into the cortex. Recordings from lateral extrastriate areas were performed using single-shank probes, which were inserted diagonally into the cortex with an angle of ~30°, in order to map the progression of RFs' centers along the probe and track the reversal of retinotopy between adjacent areas (see refs. [55,57]). The space between recording sites on each shank ranged from 25 to 200 μm; the distance between shanks (when more than one) was 200 μm; the surface of recording sites was either 177 or 413 μm². Extracellular signal was acquired with a Tucker-Davis Technologies (TDT) system 3 workstation (TDT) at a sampling rate of ~24 kHz using TDT OpenEx Software Suite. Before insertion, the probe was coated with Vybrant® DiI cell-labeling solution (Invitrogen, Oregon, USA), to allow visualizing the probe insertion track postmortem, through histological procedures.

A total of 19 rats were used for this study. Since, in some occasions, multiple recording sessions were performed from the same animal, this yielded a total of 23 sessions. After spike sorting (see below) these recordings yielded a total of 1313 well-isolated single units: V1 (510), LM (126), LI (209), LL (401). Following the application of a reproducibility filter on the neuronal responses of the units across repeated presentations of the movie stimuli (see below), the final count of single units that were further analyzed in our study was 294, of which 168 in V1 (5 different rats, for a total of 7 recording sessions), 20 in LM (4 rats, 4 sessions), 36 in LI (8 rats, 8 sessions), 70 in LL (11 rats, 12 sessions).

**Visual stimuli**. Stimuli were presented full-field to the right eye at a distance of 30 cm on a 47-inch LCD monitor (SHARP PN-E471R), with 1920 × 1080 resolution, 60-Hz refresh rate, 9-ms response time, 700-cd/m² maximum brightness, 1200:1 contrast ratio, spanning a field of view of ~120° azimuth and ~89° elevation. The stimuli were presented using Psychtoolbox[78] in MATLAB in a pseudorandom order. Between each stimulus (movie or drifting bar), a black screen was shown for at least 200 ms.

The RFs of the neurons recorded along a probe were estimated by showing drifting oriented bars over a grid of 73 locations on the stimulus display, covering the central 110° azimuth span and central-upper 70° elevation span of the total field of view of the monitor. The bars were 10° long and drifted seven times along four different directions (0°, 45°, 90°, and 135°). A spherical correction (as described in ref. [79]) was applied to each bar to compensate for shape distortions at large eccentricities. As shown in Fig. 2, mapping the inversion of the retinotopic map at each area boundary allowed identifying the visual area each neuron was recorded from[55–57].

The main stimulus set consisted of nine movies: two fast and two slow manual movies, two ratcam movies, a phase-scrambled version of one of the fast movies, and a phase-scrambled version of one of the ratcam movies (see next paragraph for details). The resolution of these movies was 720 × 1280 pixels and they were presented for 20 s at a rate of 30 frames per second (fps). An additional white-noise movie with a resolution of 45 × 80 pixels was shown at 20 fps for 20 s. All movies were converted to grayscale and were gamma-corrected offline with a lookup table calculated for the monitor used for stimulus presentation. During the course of the experiment, each movie was presented 30 times. The videos can be watched at the following link: https://osf.io/7gteq/?view_only=6300871272414e89bc2c4e7ffb03718e.

The manual movies were designed to reproduce the continuous flow of visual information typically experienced by an observer that explores an environment containing a number of distinct visual objects, placed in different locations. To this aim, we 3D-printed various geometrical objects, we painted them black or white (see examples in Fig. 3a), we placed them inside an arena and we simulated an observer roaming through such an environment by smoothly moving a hand-held camera across the arena for 20 s. The camera was moved at two different speeds to obtain slower and faster movies. The ratcam movies were intended to better simulate the visual input of a rat exploring a natural environment[49]. They were obtained by placing a small web camera (Microsoft Lifecam Cinema HD) on the head of a rat, while it was running freely inside the arena that contained some of the 3D-printed objects and another rat.

The phase-scrambled movies were obtained by performing a spatiotemporal fast Fourier transform (FFT) over the standardized 3D array of pixel values, obtained by stacking the consecutive frames of a movie. The phase spectrum was extracted and shuffled, then merged with the unaltered amplitude in such a way to

preserve the conjugate symmetry of the initial transform. An inverse FFT was then applied on the resulting array. The imaginary parts were discarded, then the original mean and standard deviation were restored. Values outside the 0–255 range were clipped in order to obtain a valid movie. The white-noise movie was generated by randomly setting each pixel in the image plane either to white or black.

**Histology**. In a few experiments (5 rats), at the end of the recording session, each animal was deeply anesthetized with an overdose of urethane (1.5 gr/kg) and perfused transcardially with phosphate buffer saline (PBS) 0.1 M, followed by 4% paraformaldehyde (PFA) in PBS 0.1 M, pH 7.4. The brain was then removed from the skull, post-fixed in 4% PFA for 24 h at 4 °C, and then immersed in cryoprotectant solution (15% w/v sucrose in PBS 0.1 M, then 30% w/v sucrose in PBS 0.1 M) for 48 h at 4 °C. The brain was finally sectioned into 30-μm-thick coronal slices using a freezing microtome (Leica SM2000R, Nussloch, Germany). Sections were mounted immediately on Superfrost Plus slides and let dry at room temperature overnight. A brief wash in distilled water was performed, to remove the excess of crystal salt sedimented on the slices, before inspecting them at the microscope. Each slice was then photographed with a digital camera adapted to a Leica microscope (Leica DM6000B-CTR6000, Nussloch, Germany), acquiring both a DiI fluorescence image (700-nm DiI filter) and a bright-field image at ×2.5 and ×10 magnification. By superimposing the fluorescence and bright-field images, we reconstructed the tilt and the AP position of the probe during the recording session (se example in Fig. 2a, left).

**Data analysis**. All analyses were performed in Python.

**Spike sorting and selection of the units included in the analyses**. Data were spike-sorted offline with the spike sorting package KlustaKwik-Phy[80] in two steps: automated spike detection, feature extraction, and expectation maximization (EM) clustering were followed by manual refinement of the sorting using a customized version of the GUI. The last step was performed by taking into account many features of the candidate clusters: (1) the distance between their centroids and their compactness in the space of the principal components of the waveforms (a key measure of goodness of spike isolation); (2) the shape of the auto- and cross-correlograms (important to decide whether to merge two clusters or not); (3) the variation, over time, of the principal component coefficients of the waveform (important to detect and take into account possible electrode drifts); and (4) the shape of the average waveform (to exclude, as artifacts, clearly nonphysiological signals). Clusters suspected to contain a mixture of one or more units were separated using the "reclustering" feature of the GUI (i.e., by rerunning the EM clustering algorithm on the spikes of these clusters only).

At the end of the manual refinement step, we further screened the resulting well-isolated single units to make sure that their firing was reproducible across repeated stimulus presentations. The selection was based on a reproducibility index that quantified, for each neuron, how reliable its responses were to its preferred frames within a movie (see definition below). This metric was chosen over other approaches that rely on computing the correlation coefficient between responses to repetitions of the same stimulus[60,81], because we wanted to avoid the inclusion of silent neurons that would have yielded high correlations due to lack of activity. To calculate the reproducibility index for a given unit and movie, we included in the metric the top 10% time bins with the highest neuronal response (given by the firing rate of the neuron in the bins). Let $X_t = \{x_k^{(t)}\}$ be the set of responses of the neuron $x_k^{(t)}$ in one of such preferred time bins $t$ (with $k$ denoting the stimulus repetition). Let $N$ be the number of repetitions. Then, the reproducibility index is calculated as $1 - \left\langle \frac{\sigma[X_t]}{\langle X_t \rangle} \right\rangle / \sqrt{N}$. The resulting metric ranges from 0 to 1[82], where 1 corresponds to ideal responses with perfectly reproducible trials. A cell was considered reproducible if, for at least one of the nine movies, the reproducibility index was 0.7. This threshold was arbitrarily set following an extensive visual inspection of raster plots from different movies to ensure that clearly responsive and reproducible cells were included. For all the analyses described in this study, the responses of these units during the presentation of the movie stimuli were converted to a spike-count representation by binning spike trains in 33ms bins.

**Characteristic timescale of the movie stimuli**. To quantify how fast the pixel-level content of the movies changed over time, we computed the following metric. Given a movie, for a given time lag $\Delta$, we computed the average correlation coefficient between all movie frame pairs separated by that lag, on a pixel-wise basis, i.e.:

$$C(\Delta) = \frac{1}{T-\Delta} \sum_{t=1}^{T-\Delta} \frac{\mathrm{Cov}[X_t, X_{t+\Delta}]}{\mathrm{Var}[X_t]\mathrm{Var}[X_{T+\Delta}]} \qquad (2)$$

where $T$ is the total number of frames in the movie, $X_t = \{x_{w,h}^{(t)}\}$ is the movie frame $t$ and expectations are taken over pixel positions, indexed by $w, h$ (for width and height). We then plotted the correlation $C(\Delta)$ as a function of $\Delta$ (Fig. 3b). To fit the decay of the correlation with $\Delta$, we considered two possible functional forms: a decaying exponential of the form $y(t) = aexp(-t/\tau)$ and a damped oscillation

function of the form $y(t) = aexp(-t/\tau)cos(\omega t + \phi)$. Fitting was performed with the basin-hopping algorithm[83] coupled with L-BFGS-B[84]. Model selection was performed independently for each fit with an Extra Sum of Squares test[85], choosing the more complex model whenever the $p$ value from the test resulted below the threshold of 0.05. The model selection procedure selected the simple exponential form in all cases. The time constant $\tau$ of the exponential decay was taken as the characteristic timescale of the stimulus.

**Characteristic timescale of stimulus-driven neuronal population responses**. To measure the characteristic timescale of neural stimulus correlations from population peri-stimulus time histograms (PSTHs) in our anesthetized rat data, we applied a very similar procedure to that described above for the visual input. We used the same expression as above (Eq. (2)) to compute $C(\Delta)$, but now we set $X_t = \{x_n^{(t)} - x_n\}$, where $x_n^{(t)}$ is the trial-averaged spike count of cell number $n$ at time frame $t$, and subtracting $\langle x_n \rangle$ centers $\{x\}\_\{n\}^{\{(t)\}}$ around its temporal average ($x_n^{(t)}$ as a function of $t$ is color-coded in Fig. 3c for the V1 and LL populations, in response to two example movies). A time constant was extracted by fitting either an exponential decay of the form $y(t) = aexp(-t/\tau)$ or a damped oscillation function of the form $y(t) = aexp(-t/\tau)cos(\omega t + \phi) + b$ for $t > 0$ (see Fig. 4a). Model selection was carried out as described in the previous section. The time constant $\tau$ of the exponential decay or of the exponential envelope of the damped oscillation was taken as the characteristic timescale of the trial-averaged population response (PSTH). The simple exponential form was selected for three of the movies in V1 and one of the movies in LL; the damped oscillation form was selected for all other movies. We note that here, as well as in the case of intrinsic timescale estimation (see below), the model selection procedure ensured that the estimated timescale matched the characteristic time over which the statistical dependence between the signal $X_t$ at time $t$ and the signal $X_{t+\tau}$ at time $t+\tau$ became small compared to its value at short lags. Crucially, the model selection procedure allowed this to hold even in presence of oscillatory patterns in the correlation functions, where the fact that the correlation goes to zero at a certain lag can hide the fact that it takes on significant negative values (indicating the persistence of a statistical relationship, albeit with flipped sign) and can further possibly come back to positive values at larger lags. We considered one population PSTH for each cortical area we recorded from, pooling together all the available sessions, as illustrated in Fig. 3c.

**Intrinsic timescale of neuronal activity**. Intrinsic timescales of neuronal activity were computed in a way that mirrors the stimulus-driven correlation timescales defined in the previous section (see Supplementary Fig. 1) and that is a simple extension of the definition given by ref. [44]. To compute $C(\Delta)$ for a given cell, we set $X_t = \{x_k^{(t)}\}$ in Eq. (1), where $x_k^{(t)}$ is the spike count of the cell at time bin $t$ in trial $k$. Following ref. [44], we then averaged $C(\Delta)$ across all units belonging to the same area before extracting a time constant (see example in Fig. 4c). The characteristic timescale of intrinsic correlation decay was computed as above for the PSTH, using the same fitting functions and the same model selection procedures. Intrinsic correlations in V1 were well fitted by simple exponential decays, while the damped exponential form was selected for fitting intrinsic correlations for all movies in extrastriate areas.

**Characteristic timescale of decoding performance**. After computing the average classifier performance $p(\Delta)$ as a function of the lag $\Delta$ (see next section), a characteristic decay time constant was extracted by the same procedure previously used to compute the timescales of the movies, of the stimulus-driven responses and of the intrinsic activity. Namely, either an exponential decay of the form $y(t) = aexp(-t/\tau) + b$ or a damped oscillation function of the form $y(t) = aexp(-t/\tau)cos(\omega t + \phi) + b$ were fitted to $p(\Delta)$ for $\Delta > 0$ (see example in Fig. 5b). Model selection was performed as described above, and the exponential decay time constant $\tau$ was extracted as the quantity of interest. The majority of pseudopopulations/movie/trial set combinations were fit with the damped-oscillatory functional form (V1: 108 out of 144; LM: 14 out of 18; LI: 17 out of 18; LL: 48 out of 54).

**Classifier analysis**. To assess the discriminability of population activity at different points in time during the presentation of a given movie, we proceeded as follows. For each recording session, we divided the 30 available recording trials in a training set of 20 trials and a validation set of 10 trials. We pooled all units that were recorded from a given area across all sessions to obtain a total number of units $N$ per area. We generated K pseudopopulations of $M < N$ units (with $M = 20$ for most of the analyses) by selecting as many random nonoverlapping subpopulations of size $M$ as possible. For instance, in V1, $K = 8$ as $N = 168$. These subpopulations were the same in the training and in the validation set. Following standard procedure when working with pseudopopulations, for each trial set (training and validation), we shuffled cell activity across trials to destroy cross-cell noise correlations.

We then considered all pairs of time bins along a trial that were separated by at least 40 time bins (i.e., by 1320 ms, as bin size was 33 ms). For each of these "reference" pairs of time bins (gray boxes in Fig. 5a), we trained a linear support vector classifier (provided by liblinear[86] via scikit-learn[87]) to discriminate population activity samples from one of the element of the pair vs. the other. The penalty hyperparameter was chosen by threefold cross-validation within the

training set, performing a grid search over candidate values $10^{-2,-1,0,1,2}$. After selecting the best value of the hyperparameter, the classifier was retrained on the full training set.

To assess the temporal stability of the population activity, the trained classifier was then tested on samples of population activity coming from different time bins than those it was trained on (orange and green boxes in Fig. 3A). More specifically, if a classifier was trained on the pair of time bins at times $t_1$ and $t_2$, it was then tested on pairs of time point at times $t_1 + \Delta$ and $t_2 + \Delta$, where $\Delta$ took on values $\{-20,\ldots,-1,0,1,\ldots,20\}$ bins*33 ms/bin. The fraction of correctly decoded trials at negative and positive time increments was then averaged to yield a classifier performance curve $p(\Delta)$ with $\Delta$ ranging from 0 to 20 bins*33 ms/bin = 660 ms. This performance curve was computed for all possible reference pairs of time bins, and averaged over all pairs. The resulting average performance decay curve (for each individual movie stimulus and pseudopopulation) was used to compute the typical timescale of self-similarity for population activity as described above. Examples of such performance decay curves are given in Fig. 5b for the case in which the test was performed on held-out trials (i.e., green boxes in Fig. 5a).

This analysis was performed separately on the training and the held-out trial sets (respectively orange and green boxes in Fig. 5a), while keeping the trained decoders fixed (see performance decay curves in Fig. 5b and Supplementary Fig. 3a). When computed on the training set, $p(0)$ corresponds to the performance of the classifier on its own training data, but $p(33 \text{ ms})$ is the performance of the classifier on an entirely new set of data, which happens to come from the same set of experimental trials as the training data. When compared to the results on the held-out set, this allowed quantifying the extent to which the self-similarity of the population activity in time was due to the stimulus-locked representational structure of the population code rather than to intrinsic, within-trial, temporal correlations in the activity of each neuron. Intuitively, population activity at time $t$ in trial $i$ can be expected to be more similar to population activity at time $t + \Delta$ in the same trial than in a different trial, precisely because of the existence of within-trial, stimulus-independent correlations in the activity of each neuron. Comparing the performance of the classifier on the training set vs. the held-out validation set, as done in Fig. 5d, allowed therefore to assess the importance of this correlational structure in enhancing the self-similarity of population activity over time (see Eq. (1)).

**Regression analysis and hypothesis testing**. Linear regression for the dependence of the timescale of stimulus-driven responses and intrinsic activity on the stimulus timescale (Fig. 4b, d), as well as for the dependence of the amount of classifier performance due to intrinsic correlations on the timescale of intrinsic activity (Fig. 5d and Supplementary Fig. 2c) was computed by ordinary least squares. In each case, residuals were not incompatible with a normal distribution (Jarque-Bera, $p > 0.05$). Regression and tests were performed with the Python package statsmodels.

The distribution of the data was less regular for the case of the classifier performance vs. the timescale of the movie (Fig. 5c and Supplementary Fig. 3b). In particular, there were a few outliers with very long decoding timescales, especially for LL, which could have biased our conclusion in favor of the hypothesis that higher areas in the hierarchy have longer processing timescales. To prevent our conclusions from relying disproportionately on these points, we performed the corresponding linear regressions using a robust estimator (Theil-Sen estimator;[88]). The confidence intervals were determined by percentile bootstrap (10000 samples;[89]) stratified by cortical area. This combination of estimator and CI determination procedure has been shown to provide reasonable coverage probability in the face of model misspecification and data contamination by outliers[88]. Following ref. [89], $p$ values were computed by subtracting the bootstrap distribution of any given parameter from its estimated value to obtain a surrogate for the null distribution. The $p$ value was then obtained as the mass of the surrogate distribution for values larger than the estimate. This yielded a one-tailed test where the null hypothesis was that the parameter value was not larger than zero. Regression and tests were performed with custom R code[90] using the boot and WRS libraries.

**Classifier analysis on hand-matched movie segments**. The results illustrated in Fig. 6 were obtained by performing a variant of the classifier analysis on a small subset of neuronal recordings, corresponding to the presentation of movie segments that were carefully matched in pairs to reproduce as much as possible the conditions under which representation invariance is typically studied with static stimuli. All movie segments were taken to be exactly 29 frames (~967 ms) long, and as described in the main text, they were all selected to show a single object, moving smoothly from the left to the right of the frame (or vice versa) over the course of the segment. The object could be black or white, leading to the existence of four categories of movie segment pairs, based on the match or mismatch between the color of the object in each element of the pair and their direction of motion: (1) opposite color, same direction; (2) opposite color, opposite direction; (3) same color, same direction; and (4) same color, opposite direction. The number of available movie segment pairs was 4, 6, 5, and 6, respectively, for the four conditions. For each segment pair and each cortical area, a classifier analysis similar to the one outlined above was performed, as follows. First of all, as above, $K$ non-overlapping neural pseudopopulations of 20 cells each were formed at random,

where $K = \text{floor}(N/20)$ and $N$ was the total number of available cells (for area LM, this just meant forming one pseudopopulation taking up all available cells). Neural activity was binned in temporal bins of duration 1/30 s, (rather than 33 ms as in the other analyses), corresponding to the exact duration of each movie frame, to ensure straightforward alignment of neural binned data and movie frames. Each of the time bins spanning the duration of either movie segment was sequentially considered as the training bin for the classifier, which was then tested on the other bins, yielding a performance vs. lag curve, where the lag could go from −28 to +28 bins depending on the identity of the training bin. For instance, when the classifier was trained on the first bin of both segments, the classifier's performance would be evaluated for lags spanning the range from 0 to +28. On the other hand, when the classifier was trained on the middle bin of both segments (bin number 15), the performance vs. lag curve would be computed for lags spanning from −14 to +14. All performance curves were then aligned and averaged (ignoring missing values), yielding a summary performance curve for lags spanning from −28 to +28. This performance curve was further "folded" by identifying positive and negative lags. The folded performance curve (spanning now from 0 to +14 bins, i.e., from 0 to ~467 ms) was then averaged across all neural pseudopopulations for the areas. Finally, this procedure was repeated ten times, each time selecting different random pseudopopulation, and the resulting curves were averaged. This yielded an average performance curve for each movie segment pair and each brain area. For each of these curves, a characteristic timescale was computed using the same curve-fitting methods described in the previous section. A final estimate of the timescale for each condition was obtained as the average timescale derived from all movie segment pairs in that condition. Statistical significance of the difference between the timescales in V1 and those of the other areas was assessed by performing a simple linear regression of the timescales of the individual movie pairs versus area identity and condition. Mathematically, this was expressed as follows:

$$\tau = \beta_0 + \beta_{LM} \cdot LM + \beta_{LI} \cdot LI + \beta_{LL} \cdot LL + \beta_D \cdot \text{Direction} + \beta_C$$
$$\cdot \text{Color} + \beta_{CD} \cdot \text{Direction} \cdot \text{Color}$$

In this expression, LM (resp. LI, LL) is an indicator variable that is 1 when the area is LM (resp LI, LL) and 0 otherwise, and Direction (resp. Color) is an indicator variable that is 1 when the movement direction (resp. object color) is the same in the two movie segments making up a pair. Therefore, for instance, $\beta_{LM}$ represents the mean difference in timescale between LM and V1 controlling for the effect of the condition (direction and color).

**Analysis of awake mouse data**. The awake mouse dataset (here and elsewhere referred to as the Allen dataset) was collected and published by the Allen Institute for Brain Science. We refer to the original publication[12] for an exhaustive description of the experimental procedures and data processing.

The Allen dataset is subdivided into individual sessions. We used the Allen Software Development Kit (Allen SDK) to extract stimulus-aligned spike timings for the relevant visual areas (LGN, V1, LM, RL, AL, PM, and AM) from the first 17 sessions included in the dataset and we pooled all units within each area. Firing rates were binned using a bin size of 33 ms, consistent with our other analyses. As detailed in "Results" section, we only considered the neural activity recorded in response to the presentation of one of the two movies in the dataset (the longest of the two, named Natural Movie 3), as the other movie (named Natural Movie 1) was composed by highly static shots, leading to an irregular structure of the temporal pixel correlations (Supplementary Fig. 4a, left) that when collapsed as a correlation function were not well described by any of the functional forms we considered (decaying exponential, damped oscillation; see Supplementary Fig. 4b, red curves). We analyzed the movie frames and the neural data using the same methods detailed above for the anesthetized rat experiment, with a few differences. First, as we only had access to one movie, we did not study the dependence of the timescales on the stimulus timescale, but we only analyzed the difference in timescales across areas. Second, all correlation functions (pixel correlations in the stimulus, as well as response and intrinsic correlations in the neural data) were well fit by simple decaying exponentials, so the damped oscillation functional form was never used.

To study the variation of the neural timescales along the visual pathway, we performed an ordinary least squares linear regression of the timescales against the anatomical score of their respective area.

**Comparison of running and resting conditions in awake mouse**. Given a recording session, we used the velocity information $v$ to identify epochs of the neuronal responses to the movies during which the mouse was either resting ($v < 1$ cm/s) or running ($v > 1$ cm/s). Each session had a number of these epochs, and by construction each rest epoch divided two adjoining running epochs. In addition, given that each movie was presented ten times within a session, the segments of the movie in which the mouse was at rest or running were different in each presentation. Since our analysis required processing the responses of each recorded neuron across multiple repeated presentations (or trials) of the same stimuli, we looked for segments of the movie where the resting (or running) epochs obtained for the various trials overlapped and we took their intersection (see Supplementary Fig. 5). The number of resting (or running) epochs shared across trials decreased as a function of the number of trials considered and, concomitantly, their duration became shorter. In principle, our comparison between the running and the resting conditions could have been affected by our specific choice in this trade-off between

number of trials and amount of available data (duration and number of the shared resting or running epochs). To avoid any such bias, we repeated our analysis for any combination of four choices of trial number $N_t$ (i.e., 3, 4, 5, and 6 trials) and five choices of the minimal duration $L$ of the epochs to be included in the analysis (i.e., 2.5, 3.3, 4.1, 5.0, and 6.6 s). For each of these combinations, we computed the stimulus-driven and intrinsic correlation functions, we fitted the exponential decays and we obtained their time constants. This procedure was repeated for all the 17 recording sessions of the Allen dataset included in our analysis, so as to obtain an estimate of the average temporal stability of neuronal activity across the seven visual areas during quiet wakefulness (i.e., during the resting epochs) as well as during active wakefulness (i.e., during the running epochs). To make sure that the time constants included in the final averages were obtained from exponential fits with similar quality, for any combination of the parameters $N_t$ and L we pooled the errors of the fits obtained across all the resting and running epochs of all areas. We then looked at the resulting, overall distribution of fit errors and only retained the time constants of those fits with an error in the lower 50th percentile. Since in general, for any given area, the numbers of resting and running epochs yielded by this selection procedure were different, we equated them by subsampling the epochs of the more populate state. This ensured that the number of time constants used to assess the neuronal dynamics of a given area was the same for the resting and running state.

**Analysis of awake rat data**. For the awake rat analyses of Supplementary Fig. 6, we used the neuronal recordings collected in ref. [60], for which we refer for a detailed description of the acquisition methodology. In brief, single-unit recordings were performed from areas V1, LI, and TO on head-fixed, awake rats, while the animals were shown natural movie stimuli. The stimulus set included 20 movies, each 5-s long, and the number of recorded neurons was 50 in V1, 53 in LI, and 52 in TO. As for the anesthetized data, in our analyses, the spike data were discretized in temporal bins 33-ms long. We discarded all neurons that did not meet the reproducibility criterion described above (under "Spike sorting and selection of the units included in the analyses"), and this reduced the number of available neurons to 44 in V1, 40 in LI, and 38 in TO. Since the structure of the data was very similar to our new recordings in anesthetized rat, we were then able to apply the same analysis pipeline to estimate stimulus timescales, response timescales, and intrinsic timescales, leading to the data reported in Supplementary Fig. 6. Three exceptions were made to this general principle: (1) since the movies were shorter, we considered a smaller range of lags over which to compute the correlations (0.5 s instead of 2 s, compare Fig. 4a, c with Supplementary Fig. 6a, c); (2) because of the particular trend typically exhibited by intrinsic correlations (with a peak at some nonzero but short lag, see Supplementary Fig. 6c), only the tail of the intrinsic correlations was considered, using only lags larger than 5 bins (=165 ms); and (3) the form of the function used for fitting the correlations was now $y(t) = a\,exp(-t/\tau)[c + cos(\omega t + \phi)] + b$. All details of the regression analysis were the same as for the anesthetized data.

**Reporting summary**. Further information on research design is available in the Nature Research Reporting Summary linked to this article.

## Data availability

The data generated in this study have been deposited on OSF under accession code doi:10.17605/OSF.IO/7GTEQ[91]. The Allen data used in this study are available online, as documented in[12]. The awake rat data from ref. [60], used in this study, are available on OSF under accession code doi:10.17605/OSF.IO/M2E6D[92]. Source data are provided with this paper.

## Code availability

The code associated to the analyses presented in this study is freely available at[91] (https://doi.org/10.17605/OSF.IO/7GTEQ).

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

## Acknowledgements

E.P. thanks Stefano Panzeri for many discussions, prior to this work, on temporally persistent neural codes and in particular on the topic of temporal noise correlations. D.Z. thanks Domenica Bueti for the discussion of temporal processing in human visual cortex. We thank Walter Vanzella for his help in setting up the first version of the decoding analysis. We thank Margherita Riggi for her help in performing the histological procedures. We thank Xiaoxuan Jia for her feedback on the content of the Allen dataset. This work was supported by: European Research Council (ERC) Consolidator Grant (D.Z., project n. 616803-LEARN2SEE); NIH Grant R01EY07977 (V.B.); NIH Grant R01NS113241 (E.P.).

## Author contributions

Conceptualization, E.P., L.S., V.B., and D.Z.; data curation, E.P., L.S, P.M., and K.V.; formal analysis, E.P., L.S., P.M., and R.C.; funding acquisition, E.P., V.B., and D.Z; investigation, L.S.; methodology, E.P., L.S., P.M., R.C., K.V., H.O.B., V.B., and D.Z.; project administration, D.Z.; resources, D.Z.; software, E.P., L.S., P.M., and R.C.; supervision, D.Z.; writing—original draft preparation, E.P. and D.Z.; writing—review and editing, E.P., L.S., P.M., K.V., H.O.B., V.B., and D.Z.

## Competing interests

The authors declare no competing interests.
