## [Peer Review File · Nature Communications]

Reviewers' Comments:

Reviewer #1:

Remarks to the Author:

The results show a lengthening of temporal scale of the stimulus-dependent response at progressively higher levels of the ventral processing stream in visual cortex of anesthetized rats. The findings further show that the stimulus-independent, i.e. intrinsic, responses become more persistent at progressively deeper downstream recording sites. The increased persistence of the temporal response is interpreted to reflect the increasing invariance of object information at higher levels of the hierarchy. Importantly, the longer temporal scale of the intrinsic response suggests that visual processing in the ventral stream relies on feedforward and recurrent mechanisms.

Finding evidence that cortico-cortical processing of visual information involves feedforward and feedback mechanisms is a welcome validation of the hierarchical model of the cortical network. However, it is debatable whether this model has been substantiated convincingly by the present study.

(1) Performing experiments in anesthetized rats is suboptimal, given numerous reports that top-down influences can only be observed in awake animals. Whether the intrinsic processing inferred in the present study reflects recurrent mechanisms is less clear than concluded in the paper. The authors should be encouraged to provide evidence that recordings in unanesthetized rats yield similar results.

(2) Another technical issue concerns the identification of areas in the assumed but insufficiently substantiated hierarchy of the ventral stream of the rat visual system. Direct evidence for the areal hierarchy would be a welcome addition.

(3) Mapping visual areas by recordings with silicon probes seems problematic due to the likely damage inflicted by multiple perpendicular penetrations. It is remarkable that using this approach, the authors were able to distinguish LM from AL, LI from LL and were able to map single unit activity in all of these areas. I am certainly aware that similar approaches for areal identification have been used and published before. This however is weak support of their validity. The authors are encouraged to provide direct anatomical evidence that the location of recordings sites were in fact LM, LI and LL.

(4) It is asserted that the V1-LM-LI-LL pathway is homologous to the ventral stream in primates. I am not aware that this argument is supported by published evidence. It would be helpful to provide the reasons for this claim or else I recommend deleting the statement.

(5) It is puzzling that deeper areas which perform invariant stimulus representations and have more persistent responses yield inferior stimulus discrimination. A more persuasive explanation is needed than that lower areas in rat encode low-level features such as contrast and illumination.

(6) To support the impact of adaptation and transient responses across the hierarchy, human fMRI data are cited. I am not convinced that comparing BOLD and spiking signals whose biological origin and time course are fundamentally different are the optimal way to bolster the argument.

Reviewer #2:

Remarks to the Author:

The lateral areas of the rodent visual system provide a compelling model to study the computations that occur in the ventral stream. In particular, in non-human primates, the ventral stream exhibits increasingly invariant representations that are thought to support object recognition despite changes in size or position. Such invariance is hypothesized to result in a degree of temporal invariance in the activity of neurons in the ventral stream as stimuli within a scene move, but do not change in nature. To determine whether there is such an increase in temporal invariance in the ventral stream, Piasini et al. presented complex naturalistic movies and artificially altered stimuli to probe the evolution of neural responses over time. They performed extracellular single-unit recordings from anesthetized rats in four different visual cortical areas (V1, LM, LI and LL) while presenting movies with different temporal and spatial structures. They then measured the timescale of neuronal activity in these areas either by correlating the trial-averaged population activity (as a measure of signal correlation) or the single-cell single-trial activity (as a measure of intrinsic correlation) as a function of time lag. In both

cases, the time scales measured in V1 were significantly shorter than those in the LM, LI and LL, suggesting an increase in temporal invariance along the hierarchy. The authors further conclude that since the signal and intrinsic correlations both increase, this suggests that the change in temporal structure is due to differences in both the feed-forward and local integration in these areas. These results are further supported by a separate analysis in which the authors use a linear classifier to distinguish time points in the movie and find that there is a greater proportion of the classifier performance attributable to the intrinsic correlation in the higher order areas. The authors interpret this result as suggesting that local circuit dynamics become increasingly important for stabilizing the neural representation of the stimulus along the cortical hierarchy. These findings of functional specialization along the ventral stream are potentially exciting and impactful for our understanding of invariant coding. However, there are a number of analytical and interpretational issues that challenge some of the authors main conclusions, and thus significantly limit my enthusiasm for this study.

Major concerns:

1. A major challenge in the study of higher order processing, such as object recognition, is to determine whether the observed temporal structure is tied to some stimulus representation rather than simply reflecting differential filtering low level features of the visual stimulus. In particular, the authors suggest that the presence of longer time-scale correlations in the neuronal data reflect higher order invariances in the stimuli, such as the continuity of the objects in the scene. The authors set themselves up nicely to dissociate these issues by including movies in which they had removed the spatial structure (the scrambled movies) and those that had no spatial structure (the white noise movies). However, the authors did not take advantage of these controls in their analyses to determine whether the increase in temporal integration is in fact linked to temporal invariance. If there is truly an increase in temporal invariance, then this should only be true for the intact movies, and not for the scrambled or white noise stimuli. Conversely, if there is also an increase in the time scale of neuronal activity during these stimuli, this simply suggests that there are different temporal filters in these areas which are not necessarily related to invariance at all. Notably, the absence of a connection to invariance (or visual processing) would significantly decrease the impact of this study.

2. One of the more potentially exciting results is the differences between the signal and intrinsic correlations and the implications for the underlying circuit mechanisms. However, it is not at all clear that these two measures can be easily segregated into feed-forward and local computations. For instance, the intrinsic correlations might reflect slow changes in brain state that are happening in V1 and propagating in a feed-forward path down the ventral stream. Conversely, the transformation of the average activity may be happening locally within each area. The authors need to either provide some compelling evidence (such as an experimental manipulation) that supports this claim, or remove it, which would significantly decrease the impact of this study.

3. The bulk of the analysis in this study rests on the fits of the temporal decay functions. The authors variously used either an exponential or a damped oscillation to fit the data and then compared the time constants of decay of the fits. However, the same time constant of decay for these two functions yield dramatically different time courses: a damped oscillation has a much steeper initial decay for the same tau. Thus, these are not directly comparable values. Indeed, in the examples, it often looks like V1 has a slower initial decay than the other areas. Were there differences in the relative proportion of best models across areas? To what degree can the chosen model explain the difference in temporal properties across areas?

Minor Concerns:

1. The measure of the percent contribution of the signal vs intrinsic correlations in Figure 3 depends on the initial performance of the model (since this is the denominator), and V1 seems to reliably have better performance. The authors need to demonstrate that this correlation of contribution cannot be explained by initial performance.

2. There were limited details on the processing and analysis of the spike time courses. What was the

bin width used and what kinds of smoothing (if any) was applied to the time courses? This is important to understand what the temporal resolution of this data is in the context of the temporal dynamics of the stimulus.

3. Overall, this paper was difficult to read, both in terms of understanding what the authors did and what they think it means. The abstract and summary at the end of the introduction in particular could be rewritten with more concrete statements about the actual experiments done and the rationale behind the conclusions in mind. Currently, without the context of having read the results they are very hard to parse. Also, below are some suggestions for improving the display:

a. The analysis performed for the intrinsic correlations was difficult to understand from the text. Perhaps a schematic, like the one used in Figure 1 to explain the signal correlation analysis, could help with this explanation.

b. It would be helpful to directly compare the time scale of the pixel correlation (in Figure 1) with the neuronal activity correlation in Figure 2 by adding the pixel correlation to this figure.

Reviewer #3:

None

Point-by-point response to the reviewers: Nature Communications manuscript NCOMMS-19-37941

We thank the reviewers for the overall appraisal of our manuscript, as well as for their comments and criticisms that were very helpful in improving our work. Below, we address each reviewer's comment (reported in red/italics font) and indicate where changes have been made in the revised version of the manuscript. For clarity, amendments to the manuscript are shown in blue in the text.

Reviewer #1

The results show a lengthening of temporal scale of the stimulus-dependent response at progressively higher levels of the ventral processing stream in visual cortex of anesthetized rats. The findings further show that the stimulus-independent, i.e. intrinsic, responses become more persistent at progressively deeper downstream recording sites. The increased persistence of the temporal response is interpreted to reflect the increasing invariance of object information at higher levels of the hierarchy. Importantly, the longer temporal scale of the intrinsic response suggests that visual processing in the ventral stream relies on feedforward and recurrent mechanisms.

Finding evidence that cortico-cortical processing of visual information involves feedforward and feedback mechanisms is a welcome validation of the hierarchical model of the cortical network. However, it is debatable whether this model has been substantiated convincingly by the present study.

We are grateful to the referee for the detailed review which, besides encouraging us to report in a more accurate and rigorous way the details of our experimental approach, motivated us to significantly enlarge the scope of the present work by including two additional datasets recorded in awake animals (mouse and rat). This led to further analyses that uncovered a striking dependence of the temporal structure of visual cortical codes on the behavioral state of the animal (quiet wakefulness vs active behavior).

We reply below to the individual points raised in the review.

(1) Awake/Anesthetized

Performing experiments in anesthetized rats is suboptimal, given numerous reports that top-down influences can only be observed in awake animals. Whether the intrinsic processing inferred in the present study reflects recurrent mechanisms is less clear than concluded in the paper. The authors should be encouraged to provide evidence that recordings in unanesthetized rats yield similar results.

To remove any confound related to anesthesia, we have performed the same analyses on analogous data recorded in behaving mice and made available by the Allen Institute. The

results of these analyses are described in a **new section** of our manuscript (**p. 16-19**) and in a **new figure (Fig. 7)**. Critically, the finding of an increase of intrinsic and stimulus-driven activity timescales along the visual cortical hierarchy is replicated in this new dataset (see **Fig. 7c-d**). Indeed, the dependence of the stimulus-driven timescale on the position of the visual areas along the hierarchy is even more evident than in our original data in anaesthetized rats. Furthermore, strikingly, the dependence of both neural timescales on the timescale of the stimulus extrapolates well from our data to this new awake dataset (see **Fig. 7e-f**).

Since, in the Allen dataset, mice were head-fixed but were allowed to run on a spin wheel, we further exploited this dataset to check the extent to which the increase of the timescales along the visual hierarchy depended on the behavioral state of the animal. We found that the hierarchical increase was much attenuated (especially for the intrinsic timescales) during quiet wakefulness (i.e., when the mouse was at rest) than during active wakefulness (i.e., when the mouse was running). These findings are described in a **new section** of our manuscript (**p. 19-21**) and in a **new figure (Fig. 8)**. The details of the analysis are explained in our **revised Methods (p. 32-33)** and in the **new Supp. Figs. 4 and 5**.

Intrigued by these results, we further explored an additional dataset that was collected, in a previous study, by K. Vinken and H. Op de Beeck (who joined us as authors of our manuscript). These recordings were performed in awake, head-fixed rats under body restraint (i.e., enforced resting) and showed no increase of the timescale along the visual cortical hierarchy (see **p. 20-21** of our revised manuscript and **Supp. Fig. 6**).

To summarize, the combined analysis of the three independent datasets included now in our study shows that the timescales of visual cortical representations undergo a very similar hierarchical increase under anesthesia and active wakefulness, and that such increase is strongly attenuated during voluntary quiet wakefulness and enforced immobilization. As commented in our **revised Discussion (p. 23-24)**, these findings are fully consistent with previous studies reporting weakened processing of the visual input by rodent visual neurons during quiet wakefulness, as compared to both active wakefulness and the anesthetized state.

Area identification and analogy to ventral stream

(2) Another technical issue concerns the identification of areas in the assumed but insufficiently substantiated hierarchy of the ventral stream of the rat visual system. Direct evidence for the areal hierarchy would be a welcome addition.

(4) It is asserted that the V1-LM-LI-LL pathway is homologous to the ventral stream in primates. I am not aware that this argument is supported by published evidence. It would be helpful to provide the reasons for this claim or else I recommend deleting the statement.

We jointly reply to the second and fourth issues raised by the reviewer, which are strictly related and concern the evidence that the lateral progression of rat visual cortical areas (V1-LM-LI-LL) forms an object processing hierarchy akin to the monkey ventral stream.

Admittedly, in our previous version of the manuscript, we did not present much details to support this statement, mainly because we have addressed this issue quite in depth in some of our previous studies (Tafazoli et al, 2017; Matteucci et al, 2019). Now, in our **revised Introduction (p. 4-5)**, we added a **dedicated paragraph**, where we explain why we chose to carry out our study on this progression of rat visual areas, and where we review some of the relevant anatomical and functional evidence supporting its homology with the monkey ventral stream. Since the Introduction was already quite long, we could not cover and discuss in depth all the relevant literature. So, we take the chance of this rebuttal letter to provide the reviewer with a more complete explanation.

There are three sources of evidence about the hierarchical, ventral-stream like organization of the V1-LM-LI-LL progression.

The first evidence comes from **lesion studies**, showing that damaging laterotemporal cortex strongly impairs rat visual pattern discrimination (Gallardo et al., 1979; McDaniel et al., 1982; Wörtwein et al., 1993; Aggleton et al., 1997; Sánchez et al., 1997; Tees, 1999).

The second evidence comes from **anatomy studies**. Based on the patterns of feedforward and feedback connections between pairs of visual cortical areas, earlier studies in the rat (Coogan and Burkhalter, 1993) and more recent studies in the mouse (Wang and Burkhalter, 2007; Wang et al., 2011, 2012; Harris et al., 2019) have shown that the visual cortical areas of these rodent species are organized according to anatomical hierarchies that are reminiscent of the monkey ventral and dorsal streams.

The third evidence comes from **functional studies** that investigated the tuning properties of rat and mouse visual areas under various visual stimulations. Imaging studies in mice have confirmed the overall consistency with the findings of the anatomy studies in terms of ventral-like processing being carried out by lateral visual cortical areas and dorsal-like processing by parietal areas (Andermann et al., 2011; Marshel et al., 2011; Glickfeld et al., 2013; Siegle et al., 2019; Froudarakis et al., 2020). More importantly, **electrophysiology studies in rats** have shown that, along the V1-LM-LI-LL progression, many key tuning and coding properties evolve as expected for a ventral-like, object-processing pathway (Vermaercke et al., 2014; Vinken et al., 2017; Tafazoli et al., 2017; Kaliukhovich and Op de Beeck, 2018; Matteucci et al., 2019). Specifically: 1) response latencies and RF sizes gradually increase; 2) information about low-level visual properties such as luminance and contrast is progressively pruned; 3) visual representations become gradually more capable of supporting decoding of object identity despite variation in object appearance (e.g., changes in size, position, luminance and orientation); 4) the sharpness of orientation tuning decreases, while the tendency of neurons to be tuned for multiple orientations increases; 5) responses to drifting gratings become progressively more phase invariant; and 6) responses become increasingly sensitive to changes in the visual environment (i.e., surprising events). Interestingly, some of these results have been recently replicated also in the mouse (Froudarakis et al., 2020).

Again, to keep the Introduction compact, at p. 4-5 we mainly report the relevant references to rat functional studies, along with references to some recent reviews of the mouse visual system. But if the reviewer will deem it necessary, we will be happy to add to our manuscript the more extended review of the relevant literature presented in this rebuttal.

Finally, we want to point out that, in the case of the Allen dataset, the rank of each mouse visual area along the anatomical hierarchy was actually quantified by an anatomical hierarchy score, defined by Harris et al. (2019) on the basis of a network analysis of the patterns of feedforward and feedback connections between each pair of visual areas. This allowed us to relate a functional property (i.e., the timescales of neuronal responses) to an anatomical rank that was independently measured on the basis of pure anatomical data.

(3) Area identification via silicon probes

Mapping visual areas by recordings with silicon probes seems problematic due to the likely damage inflicted by multiple perpendicular penetrations. It is remarkable that using this approach, the authors were able to distinguish LM from AL, LI from LL and were able to map single unit activity in all of these areas. I am certainly aware that similar approaches for areal identification have been used and published before. This however is weak support of their validity. The authors are encouraged to provide direct anatomical evidence that the location of recordings sites were in fact LM, LI and LL.

In our previous version of the manuscript, we did not explain in much detail our procedure for the functional identification of rat visual cortical areas. This is because this procedure has been already carefully explained by previous studies from our group (Tafazoli et al, 2017; Matteucci et al, 2019) and other authors (Vermaercke et al, 2014). Thanks to the comment of the reviewer, we realized that it is nevertheless useful to explain this procedure in more detail in our current study. We thus added a **new, dedicated paragraph** to our **revised Results** (p. 5-6) and a **new figure** (**Fig. 2**) to address this point.

As shown in the figure, the problem raised by the reviewer did not affect our recordings. In fact, in order to infer the identity of the areas from which our neurons were sampled, we did not perform multiple perpendicular penetrations, each aimed at a different area. Rather, we exploited the geometry of single-shank silicon probes to reach, with a single diagonal insertion, pairs or triplets of visual areas. This is made possible by the fact that V1, LM, LI and LL form an anatomical progression of adjacent cortical regions. This is now illustrated in the histological section in Fig. 2a. The image also shows how the 32 recording sites of the probe, in that recording session, were distributed among the two deepest visual areas: LI and LL. More importantly, Fig. 2a (right) shows how these two regions were identified – i.e., by mapping the receptive fields (RFs) recorded along the probe. This allowed clearly identifying the polarity of the retinotopy that corresponds to

these two regions (with the RFs moving from the temporal to the nasal aspect of the visual field in LI and doing the opposite in LL), as well as the recoding sites at which the polarity reversed, which marks the boundary between the two areas (between electrode 17 and 16). Fig. 2 also shows two additional examples of retinotopy maps and polarity reversals recorded along the silicon probes in two other recording sessions (including one where the electrode spanned three visual areas: V1, LM and LI).

We hope that this new figure and our new paragraph clearly illustrate our area identification procedure and convincingly show that the RF maps upon which this procedure rests are highly reliable (with no issue of damage being inflicted by multiple penetrations, given that, as shown in the figure, a single insertion was sufficient to target multiple areas). Finally, we want to stress that identifying visual cortical areas by mapping their retinotopy (as done in our study) is the standard procedure adopted in the vast majority of rodent vision studies, no matter whether based on electrophysiology or imaging (e.g., see Andermann et al., 2011 and Marshel et al., 2011). In particular, it is usually such functional identification procedure to provide the ground truth for the anatomical work (e.g., when the connectivity is studied by means of tracers) and not vice versa (e.g., see Wang and Burkhalter, 2007). As such, providing direct anatomical evidence of the location of the recording sites not only is not necessary, but it could not possibly yield any better accuracy in identifying the areas we recorded from than the functional mapping of the retinotopy we relied upon.

(5) Higher discriminability in V1

It is puzzling that deeper areas which perform invariant stimulus representations and have more persistent responses yield inferior stimulus discrimination. A more persuasive explanation is needed than that lower areas in rat encode low-level features such as contrast and illumination.

We agree with the reviewer that the lower decoding performance of deeper areas as compared to V1 may seem puzzling. However, this result is indeed fully consistent with the much richer representation of low-level features (especially luminance) conveyed by V1 neurons, as compared to LI and LL especially. In the previous version of our manuscript we commented on this point in the Discussion, by making reference to our previous work on this subject (Tafazoli et al. 2017). Thanks to the reviewer's criticism, we realized that this is not sufficient. Thus, in our revised manuscript, we added a **new section** to the **Results (p. 13-16)** and we performed a **new decoding analysis** (shown in the **new Fig. 6**) to better address this point.

As explained now in our revised Results, our previous work (Tafazoli et al, 2017) has extensively documented how lower areas strongly encode lower-level features of the stimulus, and how this effect has to be accounted for in order for the increase of invariance to emerge from the data. Furthermore, the finding that under carefully controlled conditions higher areas have higher linear decoding performance for object identity was also recently confirmed by an independent study in the mouse (Froudarakis et al, 2020). Even though the data in our current

study do not lend themselves to perform a delicate control such as that performed in (Tafazoli et al, 2017), we performed an additional analysis to address this point. The purpose of this analysis was to approximate as much as possible the stimulus conditions under which invariance is typically studied in static settings, and compare the absolute stimulus discrimination power across areas under varying degrees of matching of low-level stimulus features.

More specifically, we repeated the decoding analysis while focusing on segments of the natural movies where a single object was present for a sufficient amount of time (all the segments considered are 29 video frames long, i.e., approximately 967ms). Of these segments, we selected those where the object could be seen moving in a smooth trajectory either from left to right (LR) or from right to left (RL), and where the speed of the movement was approximately the same across movies, so that the positions occupied by the object in all left-to-right movie segments had a reasonable overlap with the positions of the object in the corresponding frames of all other left-to-right movie segments, and similarly for right-to-left (see Fig. 6a). Recalling that objects in the movies could be either black or white, we were therefore able to form movie pairs of four distinct types, namely all combinations of same direction/opposite direction and same color/different color. For each of these segment pair groups, we performed a specialized variant of the decoding analysis of Fig. 5 which allowed us to compare the absolute decoding performance as well as its temporal stability across areas (see our **revised Methods** at **p. 31-32** for further details). Our expectations in performing these analyses were that: (1) by restricting the movie segments to only contain one object at a time (so that object identity was guaranteed to be stable over time), the results would be more comparable to those obtained in standard analyses of coding invariance with static stimuli; and (2) that same-color movie segment pairs, being harder to distinguish based only on low-level visual features, would show a larger difference in the temporal stability of coding in V1 vs higher areas.

Fig. 6 shows the results of this analysis. Indeed, we found that, compared to the results presented in Fig. 5 for the general classifier analysis, the increase of the classifier timescales along the cortical hierarchy was much more pronounced (compare Fig. 6c and Fig. 5c). Furthermore, and **addressing directly the criticism raised by the reviewer**, we found that even though the absolute classifier performance was still the highest in V1 at low lags, extrastriate areas caught up to and even surpassed, in absolute terms, the performance of V1 at longer lags. This was particularly evident in the “same color” conditions, where all areas reached the same performance level of V1 for large enough lags (Fig. 6b). To summarize, we have shown that when stimulus conditions are controlled as well as possible given the available data to match low-level features such as size, color and position of the objects, the higher invariance of extrastriate representations emerges as a clear trend for the discrimination performance in LM, LI and LL to reach and surpass that in V1 at long lags (i.e., when the visual stimulus is allowed to change enough from the one appearing at the training temporal bin of the decoder, so as to make differences in low-level features not reliable any longer for the V1 representation).

(6) Reference to human data

To support the impact of adaptation and transient responses across the hierarchy, human fMRI data are cited. I am not convinced that comparing BOLD and spiking signals whose biological origin and time course are fundamentally different are the optimal way to bolster the argument.

We understand the point of view of the reviewer, but we want to stress that these studies are mentioned in the Introduction simply to provide a comprehensive overview of how adaptation may impact visual representations across the visual hierarchy. They are not the only evidence upon which we base and motivate our investigation, since we mention, along the human fMRI studies, several monkey and rat neurophysiology studies. As such, we believe it is appropriate to refer to this literature and we kept this section in the Introduction of our revised manuscript.

Reviewer #2

These findings of functional specialization along the ventral stream are potentially exciting and impactful for our understanding of invariant coding. However, there are a number of analytical and interpretational issues that challenge some of the authors main conclusions, and thus significantly limit my enthusiasm for this study.

We thank the referee for the constructive review, which spurred us to perform additional analyses (including on the new data we are adding in this revision of the paper), completely rewrite our introduction, and support this rewrite with a new theoretical model to better ground our intuition about the role of state- and history-dependent processing in shaping the response and intrinsic timescales of neural activity. We proceed below to address, one by one, the various points raised in the review.

Major concerns

(1) Temporal filtering

A major challenge in the study of higher order processing, such as object recognition, is to determine whether the observed temporal structure is tied to some stimulus representation rather than simply reflecting differential filtering low level features of the visual stimulus. In particular, the authors suggest that the presence of longer time-scale correlations in the neuronal data reflect higher order invariances in the stimuli, such as the continuity of the objects in the scene. The authors set themselves up nicely to dissociate these issues by including movies in which they had removed the spatial structure (the scrambled movies) and those that had no spatial structure (the white noise movies). However, the authors did not take advantage of these controls in their analyses to determine whether the increase in

temporal integration is in fact linked to temporal invariance. If there is truly an increase in temporal invariance, then this should only be true for the intact movies, and not for the scrambled or white noise stimuli. Conversely, if there is also an increase in the time scale of neuronal activity during these stimuli, this simply suggests that there are different temporal filters in these areas which are not necessarily related to invariance at all. Notably, the absence of a connection to invariance (or visual processing) would significantly decrease the impact of this study.

This is a very interesting question, and we agree that in principle a careful and systematic comparison between movies that possess or lack the spatial structure of natural image statistics could help shed some light on the relationship between invariant neural representations and temporally stable neural codes. However, unfortunately, our data did not support such analysis, as breaking down our initial dataset of 9 movies in two “natural” and “synthetic” subsets would have left us with, respectively, only 6 and 3 movies which would have been too little to reliably capture any effect. For instance, the effect of going from 9 to 3 movies in the synthetic subset, i.e. reducing the amount of available data by a factor of 3, could be compared to going from 20s-long movies to ~7s-long movies, which in **Supplementary Fig. 8a,b** is shown to greatly reduce the statistical power for the detection of stimulus-driven effects. Paired comparisons would also have been severely underpowered, as we only had two movies that were present in both an unscrambled and scrambled version in our stimulus set (‘Manual Fast 1’ and ‘Ratcam 2’).

To work around this issue, we chose instead to directly address the point as phrased by the reviewer towards the end of their remark: whether the trends we observe can be explained based on a systematic increase of the width of the temporal filters that describes the temporal integration of low-level visual features by neurons along the hierarchy. In fact, the RFs of low-level visual neurons, for which the stimulus response relationship is approximately linear, can be modeled by spatiotemporal linear kernels, whose aperture describes the spatial and temporal extent over which the visual input is integrated (Ringach, 2002; Schwartz et al., 2006; Sharpee, 2013). Such a simple description, often referred to as linear-nonlinear (LN) model, does not capture the stimulus-response relationship of higher-order neurons, which is highly nonlinear, exactly because of the computations performed to build up transformation-invariance and shape tuning (Riesenhuber and Poggio, 1999; Berkes and Wiskott, 2005; DiCarlo et al., 2012). The question is whether the hierarchical growth of the response timescales observed in our study reflects an increase of the importance this latter kind of nonlinear, invariant processes or, rather, is the result of the spatiotemporal response properties of the neurons remaining largely linear along the hierarchy, but with broader temporal kernels.

An answer to this question can be found in one of our recent studies, in which we used spatiotemporally correlated noise to infer the RF structure of visual neurons in V1 and LL by applying the spike-triggered average (STA) method (Matteucci et al., 2019). While in V1 we found many crispy, high-contrast, multi-lobed RFs with a Gabor-like structure, in LL

the STA images displaying any detectable spatial structure were fewer and those who did so were poorly defined, lower-contrast and single-lobed. This reflected an inability of STA to recover the spatiotemporal RFs of LL neurons, as expected for an area where nonlinear computations play a dominant role in establishing the stimulus–response relationship. Based on this finding and other evidence of invariant, nonlinear processing in LL gathered in (Matteucci et al., 2019), it is very unlikely that the large response timescales observed for LL in our current study are the result of linear RFs with broad temporal kernels.

In principle, a similar STA analysis would allow to directly test the extent to which LN models can account for the different response dynamics displayed by rodent visual areas in our recordings and in the Allen dataset (i.e., the pool of recordings obtained in awake mice by Siegle et al., 2019, of the Allen Institute, that we included in our revision to address the concerns of reviewer #1 about the generality of our findings in awake animals; see the **new Result section** at **p. 16-21** and the **new Fig. 7 and 8**). It would suffice to feed the movie stimuli as an input to the LN models obtained for the recorded neurons to simulate their responses and estimate the time constants of their autocorrelation functions. Unfortunately, neither in our recordings nor in those included in the Allen dataset (Siegle et al., 2019) the noise stimuli that are necessary to run a rigorous STA analysis were presented.

However, (Siegle et al., 2019) did present some stimulus conditions that allow computing at least a crude approximation of the spatiotemporal RFs of the neurons included in the Allen dataset. Namely, they presented a set of Gabor patches over a 9×9 grid of visual field locations that allow estimating at least the spatial extent of the recorded RFs (this is similar to the method we used to map the RFs in our recordings; see our **new Fig. 2**). In addition, they also presented brief, full-field flashes lasting 250 ms, and by fitting exponential decays to the spike autocorrelations functions of the resulting flash responses, they estimated a characteristic temporal scale for each of the mouse visual areas under exam, which was shown to increase along the hierarchy. While this procedure doesn't measure directly the integration time scale of the neurons seen as linear units, it can give a reasonable approximation (it can be shown that, if a neuron is a pure Poisson LN unit and it is probed with a brief stimulus pulse, the decay time constant of the resulting spike autocorrelation matches the integration time scale of the neuron). Moreover, the key need for this analysis was to have a sensible way of assigning increasing timescale values to the different areas, which this approach provided.

These considerations led to the **new analyses** presented in **our revised Discussion (p. 22)** and in the **new Supplementary Fig. 9**, where we modeled each recorded neuron in the Allen dataset as a linear filter, with the spatial kernel given by the RF profile measured with the Gabor patches and the temporal kernel estimated by the exponential fit of the autocorrelation function of the flash response, and we studied the response timescales of these linear filters to the movie stimulus used in the Allen dataset. As shown in Supplementary Fig. 9c, such response timescale did not increase monotonically as a

function of the anatomical score – a result that is very different from the one obtained for the actual responses of the Allen units to the movie (see Fig. 7c).

In conclusion, the result of this analysis, along with our previous assessment of the growth of response nonlinearity from V1 to LL (Matteucci et al., 2019), strongly suggest that the increased temporal stability of neuronal responses to natural movies across the visual hierarchy cannot be accounted for by a simple broadening of the temporal kernels of the recorded units. This supports the conclusion that the hierarchical increase of response timescales observed in our recordings and in the Allen data results from the nonlinear processes underlying the growth of shape selectivity and transformation-invariance that is expected to take place along the visual pathway.

(2) "Intrinsic" vs "local"

One of the more potentially exciting results is the differences between the signal and intrinsic correlations and the implications for the underlying circuit mechanisms. However, it is not at all clear that these two measures can be easily segregated into feed-forward and local computations. For instance, the intrinsic correlations might reflect slow changes in brain state that are happening in V1 and propagating in a feed-forward path down the ventral stream. Conversely, the transformation of the average activity may be happening locally within each area. The authors need to either provide some compelling evidence (such as an experimental manipulation) that supports this claim, or remove it, which would significantly decrease the impact of this study.

We thank the reviewer for raising this point. Indeed, our choice of words around intrinsic/feedforward seemed to have generated some confusion, and we have updated it now. Throughout the paper we had written "feedforward" to mean "directly driven by the instantaneous content of the stimulus" and "intrinsic" to indicate any component of the neural activity that could be driven by an internal state (such as the product of adaptation mechanisms, recurrent activity, and feedback signals from higher areas), and could not be seen as a simple stateless transformation of the input. Thus, the emphasis was not so much on the progressive transformation from one area to the next, but more on the idea of feedforward processing as directly (and exclusively) dependent on the stimulus at a given point in time. This was inspired by analogy of how deep neural network models of vision are often discussed: in traditional, feedforward convolutional neural networks (as well as in classic models of unsupervised learning of invariant representation, such as Slow Feature Analysis), the activity in any layer at a certain time only depends on the stimulus at that time, while in more recent models that integrate recurrent processing the activity can depend on the past history of stimuli. So in our work we investigate an essential feature of state-of-the-art models of the ventral stream (gradual buildup of representation invariance, both at the single-trial and the trial-averaged level), and we point out that under dynamic stimulus conditions this simple picture could be altered by mechanisms some of which are entirely uncontroversial (e.g., adaptation). Thus, while a departure from the naive view of feedforward buildup of invariance could be due to local neuron

processes at the single-neuron or circuit level (e.g., adaptation or recurrent intra-areal processing), or even to top-down influences, these are just examples of what is possible, and we admit that our initial wording may have misled the reader by putting too much emphasis on any one of these individual perspectives. We believe our question to be very interesting regardless of the specific feedforward/local processing angle. To resolve this confusion point we have **extensively rewritten the Introduction (p. 2-5)** to make clear that the actual question addressed by our study concerns the difference between “static”, purely instantaneous processing of the incoming visual input vs. state- and history-dependent processing (with the latter kind of processing naturally leading to intrinsic correlations of the neuronal activity over time). In addition, we added a **new figure (Fig. 1)** to clarify these concepts and a **simple theoretical model (Supplementary Text and Supplementary Fig. 8)** to make our argument more concrete.

For instance, we have exemplified in Fig. 1 how a neuron coding for a low-level, short-lived feature of the visual input may respond in a similar way to a neuron coding for a more abstract, “slower” feature but that also happens to possess an adaptive mechanism. In this case, making the response of the neuron dependent on the history of its own state (which would be seen as recurrent processing in an artificial neural network) has a strong and easily interpretable effect on its response timescale and on the way this timescale depends on the timescale of the input (Supplementary Fig. 8e, inset). This corresponds to the case raised by the reviewer of a “transformation of the average activity [that] may be happening locally within each area”, but it is also something that typically is not accounted for in popular models of the ventral stream (with the very recent exception of Vinken et al, 2020, cited in the text).

For another example, consider the effect of endowing the system with the ability to perform as a leaky integrator of its input (such integration mechanism is controlled by the parameter γ in the model described in the Supplementary text; see also Supplementary Fig. 8a). As the leak of the integrator gets smaller and smaller, any noise embedded in the system, including temporally independent noise such as that included in the model, reverberates over longer and longer times, increasing the intrinsic timescale of the neural activity (Supplementary Fig. 8h, inset). If one were then to consider a feedforward cascade of neural layers composed by units with this property, then (as we discuss in the Supplementary Text) this effect would compound along the hierarchy, and would become more and more important in determining the single-trial dynamics of the neural code. The reviewer mentions the possibility of an unspecified state fluctuation originating in V1 and propagating downstream; this would certainly give rise to temporally correlated noise in the downstream areas, but it is unclear how the resulting intrinsic timescales could increase along the hierarchy in absence of temporal processing mechanisms such as those discussed here. If anything, one could expect that the vast majority of this temporally correlated noise in V1 would be unrelated to the dimensions of the code expressing high-level concepts such as object identity (i.e., in ANN parlance, we don't expect state fluctuations in V1 to induce *adversarial* noise), and therefore that most of this noise would get squashed away by the nonlinear transformations leading to the high-level representations in later areas of the hierarchy. In absence of intrinsic processing by later areas, then (or of additional, ad-hoc sources of temporally correlated noise), the total amount of V1-originated, temporally correlated noise would then decrease along the hierarchy, leading to shorter intrinsic

timescales and to a reduced importance of intrinsic correlations for the single-trial stability of neural representations, in direct contrast to our findings (see Figures 4d and Figure 5d, also adapted in response to another point the reviewer, as well as Figure 6d, added to address an issue raised by Reviewer #1).

In conclusion, we hope we have convinced the reviewer that, while not addressing specifically the feedforward/local processing dichotomy as described in his/her remark, our study makes significant progress towards understanding the effects of a broad class of mechanisms (which include, but are not limited to, local processing mechanisms) that can alter the simple conceptual picture of the gradual invariance increase along the ventral stream when dynamic stimuli are taken into account.

(3) Fit type

The bulk of the analysis in this study rests on the fits of the temporal decay functions. The authors variously used either an exponential or a damped oscillation to fit the data and then compared the time constants of decay of the fits. However, the same time constant of decay for these two functions yield dramatically different time courses: a damped oscillation has a much steeper initial decay for the same tau. Thus, these are not directly comparable values. Indeed, in the examples, it often looks like V1 has a slower initial decay than the other areas. Were there differences in the relative proportion of best models across areas? To what degree can the chosen model explain the difference in temporal properties across areas?

We thank the reviewer for highlighting this potential source of confusion. Our answer to this point is on three levels. Briefly: (1) We maintain that our procedure is an appropriate and principled way of characterizing the most relevant timescale of a decay function (be it related to intrinsic correlations, stimulus-driven correlations or classifier performance) that could take one of two functional forms. On the other hand, (2) our disagreement on this point is immaterial, as there are no major systematic differences in the models selected by our procedure that can bias the result in favor of our hypothesis. Finally, (3) for the correlation analyses, our additional data from awake mice (Allen dataset; see the **new Result section** at **p. 16-21** and the **new Fig. 7 and 8**) yielded even stronger results than our initial data, and do not rely on multiple functional forms for fitting the correlation decays, using exclusively simple exponentials.

More in detail:

1. In all three contexts where we provided timescale estimates (intrinsic correlations, stimulus-driven correlations, classifier performance), the timescale is supposed to capture the typical time beyond which the neural signal loses "memory" of itself. In other words, the timescale τ is such that the statistical relationship between the signal $X(t)$ at time t and the signal $X(t+\tau)$ at a lag τ will tend to be "small", regardless of t (what constitutes the neural signal, as well as the way in which such "statistical relationship" is defined, differs between the three contexts). Now, if the dependence of the statistical relationship/degree of self-memory on the lag τ is well described by a simple

exponential, it is clear that the time scale of the exponential is the quantity we are after. If the function is instead a sinusoid with an exponential envelope, the initial decay could very well be "much steeper" than that of the exponential envelope, but at later lags it will overshoot zero, taking on negative values that clearly indicate the persistence of the statistical relationship (albeit with a sign flip), before bouncing back again in the positive range and so on until the lag is significantly larger than the timescale of the envelope. Because of this, when longer-lag information is available (and when the data is actually very well fit by the damped oscillatory functional form, as it is in our case), it could be misleading to only look at the initial behavior of the function when trying to draw conclusions about the timescale characterizing its convergence to zero, as it would entirely miss out on what happens after the first node of the sinusoid is reached. Therefore, looking at the timescale of the exponential envelope is in our view the most principled way of proceeding in this case.

2. Independently of point 1, the differences in proportion of best models between areas are not likely to induce a bias in our conclusion. More specifically:
 - for the analysis of the classifier performance (Fig. 5c, Supplementary Fig. 3b), the majority of pseudopopulations/movie/trial set combinations were fit with the damped-oscillatory functional form (V1: 108 out of 144; LM: 14 out of 18; LI: 17 out of 18; LL: 48 out of 54).
 - for the stimulus-driven correlations (Fig. 4b), there is no systematic trend for higher or lower areas to have a larger proportion of oscillating fits. Of the (9 movies x 4 areas=) 36 fits performed, the simple exponential decay was selected in 4 of them, 3 of which were in V1 and one in LL.
 - for the intrinsic correlations (Fig. 4d), there is indeed a systematic difference in the shape of the dependence of the correlation with lag in the data, which means that all intrinsic correlation functions in V1 are fitted with a simple exponential while all curves in all other areas are fitted with a damped oscillation. As we have argued in point 1 above, this is not an issue as our procedure is a fair and principled way of extracting the timescales. However, even if one were not to agree with our position on this point, this would not affect the conclusion that a gradient of intrinsic timescales exists along our cortical hierarchy, as LM, LI and LL are still characterized by a clear progression of increasing intrinsic timescales, and they are all fit with the same class of model.
3. As mentioned above, the new results from data recorded in awake mice also show the same trend of increasing timescales along the visual hierarchy (see Fig. 7 and 8), with the trend being even clearer than in our data for the stimulus-driven correlations. The timescales for the new data were all computed using simple exponential decays (as it provided an excellent fit), so this can be taken as an independent control showing that our conclusions are not an artifact induced by the use of two different functional forms in the fits.

We have clarified the points above by making some additions to the text (see our **revised Methods** at **p. 29-30**).

Minor Concerns

1. *The measure of the percent contribution of the signal vs intrinsic correlations in Figure 3 depends on the initial performance of the model (since this is the denominator), and V1 seems to reliably have better performance. The authors need to demonstrate that this correlation of contribution cannot be explained by initial performance.*

Thank you for bringing this point to our attention. We have verified that the relationship between intrinsic time scale and contribution of intrinsic correlations to decoding performance stays the same even if the contribution of intrinsic correlations is measured as the raw difference in decoding performance between training and testing set, rather than as a "fraction of decoding performance". We have updated the corresponding figure (**now Figure 5d** in our revised manuscript) and the main text accordingly. We have also reported raw differences in decoding performances between training and testing sets in the **new Fig. 6d**, where we repeated the decoding analysis on selected movie segments containing single objects to address a concern of Reviewer #1 (see the **new section** of the Results at **p. 13-16**).

1. *There were limited details on the processing and analysis of the spike time courses. What was the bin width used and what kinds of smoothing (if any) was applied to the time courses? This is important to understand what the temporal resolution of this data is in the context of the temporal dynamics of the stimulus.*

Thank you for pointing out that this information was not clearly presented. The bin width was 33ms, and no smoothing was applied. Our original manuscript did in fact include this information, but it was probably harder to find than necessary ("Data analysis" section: "*For all the analyses described in this study, the responses of these units during the presentation of the movie stimuli were converted to a spike-count representation by binning spike trains in 33ms bins.*"). We have now added binning information to the main text (see **p. 8** and the **legend of Fig. 3**).

1. *Overall, this paper was difficult to read, both in terms of understanding what the authors did and what they think it means. The abstract and summary at the end of the introduction in particular could be rewritten with more concrete statements about the actual experiments done and the rationale behind the conclusions in mind. Currently, without the context of having read the results they are very hard to parse. Also, below are some suggestions for improving the display:*
 1. *The analysis performed for the intrinsic correlations was difficult to understand from the text. Perhaps a schematic, like the one used in Figure 1 to explain the signal correlation analysis, could help with this explanation.*
 2. *It would be helpful to directly compare the time scale of the pixel correlation (in Figure 1) with the neuronal activity correlation in Figure 2 by adding the pixel correlation to this figure.*

Thank you for raising these points. To address them, we have:

1. **Completely reworked the Introduction (p. 2-5)** to try and be more concrete and clearer about the motivations and background of our study (this also in connection with the reviewer's second major concern, above). To this aim we also added the **new Fig. 1** with the schematic that illustrates how adaptation would cut short the responses of high-level, invariant neurons during exposure to dynamic stimuli.
2. Added a schematic to better explain the signal correlation analysis, as suggested by the reviewer (see **Supplementary Fig. 1**)
3. As suggested, overlaid the pixel correlation curve on the signal correlations in our **revised Fig. 3a**.

Reviewers' Comments:

Reviewer #1:

Remarks to the Author:

This is a largely successful revision. With the addition of the of the Allen mouse data the results are now more tightly linked to the network function in awake animals. Moreover the proposed temporal processing hierarchy can now be compared to a structural hierarchy score, which is lacking in the rat model. The downside of the revision is that it made the paper longer. This could be remedied by trimming the excessive wordiness throughout the text which distracts of the key message of the paper. One place to cut is repeating the results in the discussion.

Reviewer #2:

Remarks to the Author:

In response to the previous round of reviews, the authors have revised all sections of the manuscript, including new data, analysis and modeling. The data that the authors have provided on the identification of areas is particularly convincing, and use of the Allen Institute dataset was resourceful. Unfortunately, there are still major issues with this study:

- It is difficult to understand what the data/analyses tell us about the overall questions about temporal integration and object invariance.
- There remain concerns about interpretation of the data.
- The insights provided are largely phenomenological, with limited insight into the biology and/or mechanisms.

As a result, in its current state the manuscript is unlikely to enrich the broader audience it proposes to target.

Major concerns:

1. My first major concern remains that "the observed temporal structure is tied to some stimulus representation rather than simply reflecting differential filtering low level features of the visual stimulus." The argument that the temporal structure should reflect some higher-order processing in higher-order areas is implicit to the framework proposed in figure 1, where neurons in higher areas track the rat's head rather than edges. However, I am still not convinced that the authors have demonstrated that there is a link between the longer observed timescales and invariant visual processing. Rather than addressing this problem head-on with their scrambled movies, the authors instead focused on one suggested alternative, that the temporal kernel might be different across areas. Indeed, they show from responses to transient stimuli recorded across areas by the Allen Institute that there are longer static timescales that correlate strongly with the timescales of correlation from the natural movies. Yet, they then rule out the possible contribution of these longer impulse responses as they do not generate longer timescales of correlations when responses to movies are reconstructed using a linear model. However, there is no clear validation of their model; further in a different part of the manuscript the authors argue that the receptive fields of neurons in the higher-order areas are not linear, and thus it is unclear what justification the authors have for using a linear model. Thus, I am not convinced that the observed differences in integration are not important for the timescale of visual responses.

The authors also tangentially tried to address this concern by extracting four 1s clips of movies in which single objects of different luminance are moving in different directions. They find that neurons in higher order areas are subtly better able to discriminate movies at longer latencies in some conditions. However, it's not obvious why some this happens in some conditions (opposite color/same direction and same color/opposite direction) and not others, or to what degree differences in receptive field size would be sufficient to explain these results. In addition, this analysis is restricted to 500 ms delays, and therefore the decays have not plateaued and the fits are therefore extrapolated- often for

taus significantly longer than the clip. Moreover, the reason that the authors declined to analyze the scrambled datasets was that there weren't enough of them for sufficient power; thus, it is confusing how these four 1 second clips give sufficient power.

Finally, the inclusion of new data showing that this invariance does not occur during quiet wakefulness, and is not unique to the ventral visual pathway, suggest that this likely has little to do with invariance of object representations.

2. My second major concern relates to the confusion of what was meant by feedforward versus intrinsic contributions. The authors have now made it clear that they meant for this to relate more to the construction of neural networks than in a biological sense. While this is a reasonable framework, I find it more suited to a specialized computational audience.

I also fail to see their claims for "significant progress towards understanding the effects of a broad class of mechanisms". The only real effort to address mechanisms comes through their added conceptual model in which they suggest the relative contribution of adaptation and leaky integration to both stimulus-dependent and intrinsic timescales. The ideas proposed are indeed interesting- especially the non-intuitive relationship between adaptation and intrinsic timescales which aren't apparent from the predictions in Figure 1. However, they are relegated to the supplement, only integrated in the discussion, and presented with little clarity. Thus, these details are not likely to be appreciated by a general audience. Moreover, there is no effort to determine whether these features actually support the observations.

Further, there is no effort to determine the contribution of changes in spatial integration, or the circuits that control spatial integration, to the observed temporal integration. Spatial integration is known to be differentially regulated in aroused/anesthetized conditions as compared to quiet wakefulness, and thus is a likely candidate to explain some of the observations. Indeed, it is concerning that the effect is only present when receptive fields might be of their larger sizes (as in the anesthetized/locomoting conditions), suggesting that much of the non-intrinsic effects could be explained by differences in receptive field size across the hierarchy. However, it would potentially be interesting if the intrinsic mechanisms might also be modulated by these same circuit mechanisms. However, this is not addressed.

Thus, while the authors have potentially interesting results and ideas, it is not clear what mechanistic insights the authors have actually provided.

3. The authors have reasonably addressed my concerns about the fits. However, I still find it a strange practice to lump all of the fits together given that the temporal integration of an exponential and a damped oscillation are fundamentally different, with the damped oscillation having a short initial window followed by a prolonged correlation. This is especially true given that the authors model suggests that the damped oscillation might be a key signature of the network state in their conceptual model.

4. The authors have improved the clarity of their abstract and introduction. However, the manuscript remains incredibly difficult to read. It suffers simultaneously from a lack of detail and clarity for how specific analyses were validated as well as an abundance of explanation for why certain routes were not taken. This made it difficult to navigate and determine which were the most important findings and how they contribute to the novelty and overall message of the manuscript.

Minor concerns:

Why do all of the linear regressions have exactly the same slope? This seems unlikely from chance, and suggests that there is some constraint on the fit that biases towards having unique intercepts. It

looks reasonable for data in Figure 4, but the V1 data in figure 5C have shorter timescales of discriminability for movies with short timescales, but similar timescales for longer timescale movies, and thus should have a steeper slope for the regression.

Nature Communications manuscript NCOMMS-19-37941A

(The comments of the reviewer are shown in **black**; our replies are shown in **blue**)

Response to Reviewer 1

Reviewer #1 (Remarks to the Author):

This is a largely successful revision. With the addition of the of the Allen mouse data the results are now more tightly linked to the network function in awake animals. Moreover the proposed temporal processing hierarchy can now be compared to a structural hierarchy score, which is lacking in the rat model. The downside of the revision is that it made the paper longer. This could be remedied by trimming the excessive wordiness throughout the text which distracts of the key message of the paper. One place to cut is repeating the results in the discussion.

We thank the reviewer for the supportive remarks. Following their advice, we have made some cuts to the text (especially in the Discussion). We would be happy to further condense the paper if the Editor will deem it useful for the final version.

Detailed response to Reviewer 2

Reviewer #2 (Remarks to the Author):

In response to the previous round of reviews, the authors have revised all sections of the manuscript, including new data, analysis and modeling. The data that the authors have provided on the identification of areas is particularly convincing, and use of the Allen Institute dataset was resourceful. Unfortunately, there are still major issues with this study:

- It is difficult to understand what the data/analyses tell us about the overall questions about temporal integration and object invariance.
- There remain concerns about interpretation of the data.
- The insights provided are largely phenomenological, with limited insight into the biology and/or mechanisms.

As a result, in its current state the manuscript is unlikely to enrich the broader audience it proposes to target.

We strongly dissent with the first two criticisms above, with the dismissive view of phenomenological studies implied by the third criticism, and with the ensuing conclusion that our work is unlikely to enrich a broad audience within the neuroscience community. We provide detailed, in-depth replies to each of these criticisms below, by giving point-by-point responses to the more specific comments by the reviewer. Here, we quickly summarize our views with regard to the three bullet points above:

- Our analyses very clearly and convincingly show that a systematic increase of both response and intrinsic timescales takes place along two visual processing hierarchies in two different rodent species. Our decoding analyses, in particular, firmly establish that, in the rat, such hierarchical growth has the functional implication of affording progressively larger degrees of perceptual constancy (i.e., invariance to object transformations);

- No ambiguities exist in the interpretation of our findings with regard to the conclusions summarized in the previous point - because the alternative, trivial explanation proposed by the reviewer cannot possibly account for our results;
- We fully reject the notion that only mechanistic studies yield important insights into brain processes. Phenomenological and functional studies of neural processing (as the one presented in our manuscript) have been and still are an essential tool to understand neuronal codes in the brain. They deserve the same respect and consideration as mechanistic studies, especially when they investigate complex, poorly understood and under-studied phenomena, such as the unfolding of visual representations under dynamic stimulation (as done in our work).

Major concerns:

First concern

1. My first major concern remains that “the observed temporal structure is tied to some stimulus representation rather than simply reflecting differential filtering low level features of the visual stimulus.” The argument that the temporal structure should reflect some higher-order processing in higher-order areas is implicit to the framework proposed in figure 1, where neurons in higher areas track the rat’s head rather than edges. However, I am still not convinced that the authors have demonstrated that there is a link between the longer observed timescales and invariant visual processing. Rather than addressing this problem head-on with their scrambled movies, the authors instead focused on one suggested alternative, that the temporal kernel might be different across areas. Indeed, they show from responses to transient stimuli recorded across areas by the Allen Institute that there are longer static timescales that correlate strongly with the timescales of correlation from the natural movies. Yet, they then rule out the possible contribution of these longer impulse responses as they do not generate longer timescales of correlations when responses to movies are reconstructed using a linear model. However, there is no clear validation of their model; further in a different part of the manuscript the authors argue that the receptive fields of neurons in the higher-order areas are not linear, and thus it is unclear what justification the authors have for using a linear model. Thus, I am not convinced that the observed differences in integration are not important for the timescale of visual responses.

Linear Filtering model

It is important here to distinguish between the criticism and the approach that is best suited to address it.

As we acknowledged in our previous reply, the criticism is a valid one. Indeed, invariant coding is not the only hypothesis that predicts increasing timescales along the hierarchy; a simple scenario that sees visual neurons as low-pass filters or temporal integrators with increasingly broad temporal kernels would also yield similar predictions. In the following, we will call this scenario the “filter model”. We take the filter model to be the alternative null hypothesis proposed by the reviewer, since:

1. It is a sensible approach that rests on a very popular conceptual view of sensory neurons (as well as a standard approach in circuit theory, signal processing, and neural coding). The idea that a visual neuron can be meaningfully described by a temporal kernel or an impulse response function is a reasonable first approximation to consider, and it is the temporal analogue of the idea that that same neuron can be meaningfully described by a receptive field.
2. In either the first or the second sets of comments by the reviewer, there are no specific details that would suggest that they had something else in mind; there is no concrete description of an

alternative hypothesis, but only generic references to “temporal filters” that may be “different” across areas.

3. If we had access to data recorded with **spatially-but-not-temporally** scrambled movies, as the reviewer seems to think we do (but see below for a detailed discussion of how we believe the reviewer misunderstands what our scrambled movies are), comparing the response timescales to scrambled and intact movies would indeed allow us to rule out the filter model, if the increase in timescale is really due to invariance. This would happen very clearly, exactly as described by the reviewer. So, the reviewer’s insistence that looking at scrambled movies would be the superior (more “head-on”) approach lends more support to the interpretation that the alternative model proposed by them is what we called the filter model.
4. Finally, as the reviewer also picks up on, the temporal kernels measured in the Allen study do in fact increase along the hierarchy, and therefore it is a good idea to check if a simple conceptual model like the one just outlined could be sufficient to explain our observations.

Given the above, we will now respond to the two specific points raised here: that (1) comparing scrambled and natural movies in our data would have been the most direct, or “head-on” way of falsifying the filter model, and that (2) the linear filter analysis presented in Supplementary Figure 9 is somehow not well validated.

Asserting that we are not addressing the issue “head on” implies that looking at the scrambled movies would somehow be the most direct way of responding to the criticism that the difference in slowness is due to differential filtering of low-level features. The argument is that if the observed difference in slowness is due to invariant processing, then this difference should disappear in the scrambled movies, as the scrambled movies lack the structures that more invariant neurons and circuits are supposedly coding invariantly for. **This is factually incorrect.** Indeed, Matteucci and Zoccolan (2020) have already shown that the amount of structure in correlated-noise movies (analogous to, and possibly even further from natural movies than, the phase-scrambled movies used here) is enough to elicit different time courses in classes of neurons that have been identified, through independent means, as being more or less invariant to spatial transformations (i.e., simple vs complex cells in V1). So, although they surely possess a simpler statistical structure than natural movies, the scrambled movies would not be a good control for the issue raised by the reviewer (even if we had enough of them to guarantee sufficient statistical power), as what structure they retain is enough to engage invariant processing and elicit longer coding timescales in complex cells (see Fig. 4 in Matteucci and Zoccolan, 2020).

To understand how this can be, perhaps it can be useful to recall the properties of phase-scrambled movies. It is **incorrect** to say that Fourier phase-scrambling results in “remov[ing] the spatial structure” of natural movies, as stated by the reviewer in their previous report. What Fourier phase-scrambling does is to reduce the spatiotemporal structure of the original movie, while ensuring a good frame-by-frame match between original and scrambled movie with respect to average pixel intensity, contrast, changes in pixel intensity across successive frames, and spatial power spectrum (Vinken et al, 2014). Thus, **the scrambled movies do contain a rich spatiotemporal structure** (they are not at all similar to white noise) – simply, it is not the same structure of the original movies. This can be appreciated visually, by looking at examples of Fourier phase-scrambled images used, for instance, in recent studies of rodent vision. Very clear examples can be found (e.g.) in Vinken et al, 2014 (Fig. 1 and 2), Vinken et al, 2016 (Fig. 1) or Froudarakis et al, 2014 (Fig. 1). The images in these examples feature rich spatial structures, with dark and bright blobs/patches with various shapes and spatial frequencies. Moreover, **the phase-scrambled movies are not only correlated in space but also in time.** This means that if a given visual feature is present in a given frame (frame t) of the scrambled movie, it will not abruptly disappear in the following frame (frame $t+1$). Rather, it will smoothly transform. Thus, a higher-order, invariant neuron will have the chance to keep responding to that feature at frame $t+1$, if it did so at frame t . This in turn can be

immediately appreciated by comparing the pixel autocorrelation functions in Figure 3b between natural movies, their scrambled counterparts, and white noise, or by watching the movies themselves (Supplementary Material).

After stating that our method is not as “head-on” as looking at the scrambled movies, the reviewer goes on to address some specific criticism to the additional analyses we included in the revised paper. The chief concern here seems to be that the linear filtering model we considered is not well justified. But this is very confusing: as we have discussed at length above, **some sort of linear filtering seems to be the only coherent interpretation of what the reviewer themselves proposed in their initial criticism**. A temporal kernel can always be estimated by examining the response to a transient input (as done in the Allen paper), but it will only be useful to predict the behavior of the system if the system is approximately linear. So the fact that the timescales of the kernels increases along the hierarchy, but a linear filtering model that uses these kernels (i.e., the best available linear filters that describe the neurons in the temporal dimension) is entirely incapable of reproducing the observed increase in activity timescales is strong evidence for the fact that this phenomenon must be due to nonlinear processing, or in other words, that the simple conceptual picture proposed by the reviewer does not hold.

On the other hand, arguing, as we have done in our manuscript, that nonlinear processing is important in rodent visual hierarchies is hardly surprising. Indeed, we performed a dedicated study to test exactly this phenomenon in the rat (Matteucci et al, 2019), and Siegle et al (2021) have applied our approach to test the same idea in the mouse. These studies have convincingly shown that nonlinear computations play an increasingly dominant role in determining the stimulus–response relationship along both visual hierarchies. What does this imply for our analysis? Very simply, it means that it was not strictly necessary to carry it out, as it was highly unlikely that simple, across-areas variations of temporal filtering properties could have explained our findings. In other words, the concern that the observed hierarchical increase of temporal scales may be trivially explained by a widening of some temporal filters was, in fact, an overconcern. Still, in our revised text we double checked that this null hypothesis scenario could be ruled out. Not only because it was asked by the reviewer, but because we deem it essential to test the solidity of our conclusions in the most systematic, exhaustive and thorough way. But this is merely a sanity check, not a critical analysis upon which the validity of our whole study hinges. This is the reason why this analysis was presented and discussed rather briefly in the Discussion and illustrated in a supplementary figure.

So, to summarize, **the reviewer challenged our interpretation of the data by providing an alternative conceptual model (differential filtering of low-level features), and we responded by implementing it and showing that it does not account for the observed increase in time scales**. The model can't be “validat[ed]” any better, because it is not a good model of the observed phenomena; but this is not a problem of our analysis – it is only evidence that the concern raised by the reviewer can be laid to rest. It is backward on the reviewer's part to claim that our response is unsatisfactory because we did not validate sufficiently the model they proposed, showing it to be invalid. Therefore, we believe we have successfully responded to the criticism, and we cannot see how this point can be valid ground for dismissing our work.

Hand-matched segment analysis

The authors also tangentially tried to address this concern by extracting four 1s clips of movies in which single objects of different luminance are moving in different directions. They find that neurons in higher order areas are subtly better able to discriminate movies at longer latencies in some conditions. However, it's not obvious why some this happens in some conditions (opposite color/same direction and same color/opposite direction) and not others, or to what degree differences in receptive field size would be sufficient to explain these results. In addition, this analysis is restricted to 500 ms delays, and therefore the decays have not plateaued and the fits are therefore extrapolated- often for taus significantly longer

than the clip. Moreover, the reason that the authors declined to analyze the scrambled datasets was that there weren't enough of them for sufficient power; thus, it is confusing how these four 1 second clips give sufficient power.

Let's clarify the meaning of the new analysis shown in Fig. 6. First, it was not meant to “tangentially” address the concern of the reviewer about temporal filtering. There was no need for that, as this concern was addressed and settled directly by the linear filtering analysis discussed above and presented in Supplementary Figure 9. The analysis of Fig. 6 was meant to address a concern of reviewer #1, who, understandably, wondered why, in terms of absolute magnitude of decoding performance, V1 was superior to downstream lateral areas in the analysis shown in Fig. 5. But **reviewer #2 is right when they say that Fig. 6 provides additional evidence against the possible role of differential temporal filtering across areas in establishing the hierarchical increase of temporal scales observed in our study** (and we thank them for this insight). In fact, the most striking trend observable in Fig. 6 is how much slower responses in higher-order areas (i.e., LI and LL) are, as compared to V1 and LM – the green (V1) and blue (LM) curves decay very abruptly over time, while the purple (LI) and red (LL) curves are almost flat (Fig. 6b). As a result, the time constants of the decays are much larger in LI and LL, as compared to V1 and LM (Fig. 6c). But more important, for the issue raised by the reviewer, is the comparison with the sister analysis performed in Fig. 5. In Fig. 6c, the time constants for LI and LL are up to five times larger than those obtained for the same areas in Fig. 5c. Crucially, the difference between the two analyses is that in Fig. 6 we analyze movie segments containing single objects and this allows the invariance of the representations to play a much stronger role in determining the temporal stability of the responses, as compared to Fig. 5 (where there is no constraint on whether and how many times object identity changes during the course of the analyzed epoch). Given that the neurons in Fig. 5 and 6 are exactly the same and have, therefore, identical time filtering properties, we can conclude that **the 5-fold increase of slowness from Fig. 5 to Fig. 6 is driven by the non-linear processes underlying invariance** – the processes that allow high-order neurons to tolerate identity-preserving transformations of visual objects. Thus, **Fig. 6 provides indeed additional, strong (not “tangential”) evidence that the larger temporal stability of object representations in LI and LL is based on non-linear, truly invariant processing**, and not on mere differences in the width of temporal filters. **We have added a paragraph to discuss this point in the Discussion of our revised manuscript (p. 22).**

Other criticisms

The other criticisms of the reviewer to this analysis are more technical and we reply to each of them below.

First, the goal of this analysis was to make it clear that, given the very rich amount of (low-level) visual information encoded by V1 neurons (please refer to Tafazoli et al., 2017, Fig. 3B – the amount of visual information in V1 is at least twice as large as in LL), and given the smooth nature of the movies (whatever visual feature a neuron may encode, that feature will not change abruptly from one frame to the next), it cannot be possibly expected for higher-order areas (LI and LL) to yield larger classification performance than lower-order areas (V1 and LM). So, it is not surprising that V1 will yield larger absolute classification accuracy than LI and LL. **What actually matters, in order to compare the invariance afforded by the different areas, is not absolute accuracy but the relative change of accuracy over time.** In fact, **this is the neuronal correlate of perceptual constancy** (which is the functional implication of invariance) when representations are tested with dynamic stimuli. A representation that allows accurate discrimination of two visual frames, but that, after a few hundreds of ms of temporal evolution of the dynamic scenes, has lost a large fraction of its initial discriminatory power does not support perceptual constancy. This is the case of V1 and LM. A representation that has a lower discrimination power to start with but maintains it (more invariantly) over time affords instead much larger perceptual constancy. This is what LI and LL do. And this is what emerges more clearly from the analysis in Fig. 6, after fixing object identity. The fact that, at large time lags, LI and LL performance slightly overcomes V1 performance also

in absolute terms simply adds additional strength to the conclusion that, under the most challenging identity-preserving transformations, the invariance of higher-order areas emerges very clearly.

On the other hand, the question why such takeover of absolute performance at long time lags takes place for some conditions and not for others is only a minor detail, and not relevant to the broader points discussed here. In our manuscript we already explain at length why we expect this takeover to be quite strong for some specific comparison (same color & opposite direction). This explanation could of course always be improved and clarified, but again, the core result here is that for all conditions we have the same qualitative trends: a sharp, fast drop of performance for V1 and LM, and a way more gradual decrease (almost flat) for LI and LL.

As to the reviewer's question of "to what degree differences in receptive field size would be sufficient to explain these results", the answer is twofold. On the one hand, RF size obviously matters. In general, it does help increase the invariance of the representation. If it didn't, the increase of RF size would not be a common feature shared by all known visual processing hierarchies (i.e., both the ventral and dorsal streams) and phylogenetically preserved across mammals (found both in primates and rodents). So, **dismissing RF size increase as a property that could trivially explain our results (as the reviewer seems to imply) is a misconception.** On the other hand, **large RFs alone are not at all enough to support invariant discrimination.** We have demonstrated this in a computational study a few years ago (Li, Cox, Zoccolan and DiCarlo, 2009). Specifically, we have shown that, when object representations are analyzed using linear classifiers (as done in our present manuscript), the key property to achieve invariance is the preservation of the rank-order of object selectivity across transformations. In a more recent study (Tafazoli et al, 2017), we have explicitly verified that this property increases along the rat V1→LM→LI→LL pathway, and, in fact, LL affords larger invariance than V1 also when RF sizes are artificially matched. In conclusion, because of these previous studies, we can say that **the larger temporal stability of the decoding performance observed for LL and LI cannot possibly be accounted for by RF size increase alone**, if this was the concern of the reviewer. **The nature of the analysis itself (based on linear classifiers) requires preservation of rank-order tuning, in addition to large RFs, for invariance to emerge.** This is a matter already settled in the ventral stream literature.

As to the fact that our "analysis is restricted to 500 ms delays, and therefore the decays have not plateaued and the fits are therefore extrapolated" we do not really see what the problem is. Fitting procedures are often applied in such scenarios. The important thing is for the fit to be carried out rigorously, as we have done. Even more importantly, **the trends shown in Fig. 6b speak for themselves.** It is not a matter of relying on exotic analytical tools to reveal minute differences of time constants in the decays of the different areas. **Differences are huge, macroscopic**, and they can be very easily appreciated with the naked eye. Trends in LI and LL are way more stable than those in V1 and LM, and **the time constants resulting from our fits (shown in fig. 6c) simply capture with fidelity these macroscopic differences.**

Finally, the reviewer states "the reason that the authors declined to analyze the scrambled datasets was that there weren't enough of them for sufficient power; thus, it is confusing how these four 1 second clips give sufficient power." The sense in which power is insufficient (even if performing the analysis would be a good idea to start with - see discussion above) is that only two of our movies are available in scrambled form, and therefore it would not have been meaningful to compare the result of our core analysis (Fig. 4 and 5) in a paired scrambled-vs-non-scrambled way. With that mention of statistical power, we were not referring to the shorter-clip analysis, which, by using only a carefully selected subset of the data, has different characteristics – it allows one to draw conclusions about what happens when only one object is shown. It is not possible to replicate this analysis on the scrambled movies because there is no meaningful way of selecting movie segments based on object identity. Finally, we have noticed that the reviewer must have thought that **the four example pairs** of movie segments shown in Fig. 6a were the only ones used in our analysis. This is not the case, as can be understood by: 1) reading the legend of

the figure, where we say upfront that these are in fact examples; 2) reading the relevant section of the Methods (page 32), where we explain that **the single-object movie segments were combined to obtain 4, 6, 5 and 6 pairs** respectively for the four conditions tested the figure (as also appreciable by looking at the scatter plots of Fig. 6c); and 3) reading the relevant section of the Results, where we said that our analysis is based on 21 such movie segments (in fact - this was a typo, for which we apologize; there are 21 pairs, not 21 segments, as we report now in our revised manuscript at **p. 15**; however, this should have nevertheless conveyed the idea that the actual number of movie segments was larger than those shown in figure 6a).

Invariance in quiet wakefulness and other visual areas

Finally, the inclusion of new data showing that this invariance does not occur during quiet wakefulness, and is not unique to the ventral visual pathway, suggest that this likely has little to do with invariance of object representations.

It is very hard for us to understand what elements support this speculation of the reviewer without further elaboration on their side. We can nevertheless explain why we think this interpretation of our findings is inconsistent with the functional role of the ventral stream (and of visual processing streams in general) and with the current understanding of the different processing performed by the rodent visual system under active and passive wakefulness.

First, our data show that, when mice are quiet/still, the hierarchal increase of both response and intrinsic time scales is strongly attenuated (but not fully abolished). As explained in our manuscript, this is consistent with previous reports of “attenuated” processing of visual features under passive wakefulness. In the literature, the current interpretation of such phenomenon is that the visual system becomes disengaged with the visual input, when the rodent is not actively interacting with the environment. Our finding is fully consistent with and adds strength to this hypothesis. If the increase of time scales has a functional meaning (and our decoding analyses show that it does: it supports perceptual constancy), then it makes sense that the amount of computational and metabolic resources allocated to sustain this function (e.g., to contrast the effect of adaptation through attentive mechanisms) is strongly reduced when the animal needs it the least (i.e., during quiet wakefulness). Following this reasoning, one would expect the visual processing hierarchy to be maximally disengaged (and thus the hierarchical increase maximally attenuated) in animals that are habituated to fully passive experiments under body- and head-restraint – which is what we found with the third dataset analyzed in our study (head-fixed rats). In summary, **if any interpretation can be derived from the active vs. passive wakefulness comparison is one of strong support for the functional role of the processing hierarchy in mediating perceptual constancy**. If we had found the opposite result (i.e., a sharper hierarchical increase under quiet wakefulness), that would have been highly problematic in terms of interpretation. It would have been in full disagreement with the findings of previous studies and with the resulting hypothesis that cortex becomes disengaged under passive wakefulness. And would have cast doubts on the actual functional role of the visual hierarchy in supporting perceptual constancy when rodents need it the most (i.e., during active engagement with the environment).

With their second criticism (that “invariance [... being] not unique to the ventral visual pathway, suggest that [...] has little to do with invariance of object representations”) the reviewer seems puzzled by the fact that the increase of timescales takes place not only across the rat object-processing pathway, but also across the whole visual anatomical hierarchy of the mouse, which likely encompasses both ventral and dorsal areas. This finding, however, does not make any weaker our conclusion that such a growth, in the case of rat ventral pathway, is the result of an increase of the invariance of object representations.

Rather, it implies that **the extension of temporal processing scales is a general, common feature of any visual processing stream**, ventral and dorsal alike (and most likely of any sensory processing stream). Such phenomenon should not be interpreted to suggest that temporal scales increase regardless of invariance. The opposite is true; as we have explained at length in our Introduction, **temporal persistency directly implies invariance in case of dynamic stimuli**, unless other processes, such as adaptation, intervene to contrast invariance. Thus, what the Allen data show is that, in general, across a processing hierarchy, invariance (in the form of temporal persistence) increases. And this makes sense, because **the ultimate goal of any sensory processing hierarchy** (not only a visual one, not only the ventral stream) **is to achieve the perceptual constancy** that is necessary to guide behavior. This applies not only to object representations, but also to color and motion representations. For instance, the so-called “pattern cells” along the monkey dorsal stream acquire the ability to encode motion direction in a way that is invariant to shape changes (i.e., they respond similarly to gratings, plaids and even more complex visual patterns, as long as they move in the same direction). This **invariance is crucial to support perceptual constancy of motion direction**. Think of an animated agent that continually reconfigures its body while walking, running or flying along a given direction (e.g., because of the displacement of legs or wings). A V1 representation would see an ever-changing cacophony of disparate, local motion directions, varying quickly over time. An MT or MST representation would report instead a global, unique and temporally stable motion direction. Obviously, no vision scientist would conclude from this that temporal persistence has a lot to do with invariance of motion representations and little to do with invariance of object representations, or vice versa. Because **temporal persistence, as a manifestation of perceptual constancy under dynamic stimulation, has a lot to do with both motion and object representations – with both ventral and dorsal processing**. This is what the growth of temporal scales across both the rat ventral stream and the mouse visual hierarchy shows in our study. And this is a finding that adds strength and generality to our study. **We have added a paragraph to discuss this point in the Discussion of our revised manuscript (p. 23).**

Second concern

2. My second major concern relates to the confusion of what was meant by feedforward versus intrinsic contributions. The authors have now made it clear that they meant for this to relate more to the construction of neural networks than in a biological sense. While this is a reasonable framework, I find it more suited to a specialized computational audience.

We are very puzzled by this statement of the reviewer. We do not really understand how the very articulate and accurate explanation of **what response and intrinsic time scales are in the brain** (see our revised Introduction) can possibly bring a reader to think that these phenomena “relate more to the construction of neural networks than in a biological sense”, leading to the conclusion that this framework is “more suited to a specialized computational audience”. In our Introduction, we dedicate four full paragraphs to explain: 1) what invariance is, when measured with static images **in the brain**; 2) how invariance is supposed to translate into temporal persistence, when measured with dynamic scenes **in the brain**; 3) how such equivalence between invariance and temporal stability can possibly be undone by activity-dependent processes known to be at work **in the brain** (e.g., adaptation and predictive signals); and 4) how other kinds of activity-dependent processes known to be at work **in the brain** (captured by intrinsic correlations) can instead extend the temporal stability of visual representations. **This is all done with reference to the brain, and specifically to the visual system literature**. Modeling work is also cited, but it is not at all necessary to frame or understand the hypotheses tested in our study, which is an empirical investigation of important, yet poorly understood, neurophysiological processes and their functional implications. Reference to feedforward neural networks and to models of neuronal adaptation

are included only to provide also a theoretical framework that helps picturing and understanding the possible mechanisms underlying such biological processes observed in the brain. Thus, **reference to computational work enriches and clarifies the broader theoretical context of our study but is not at all necessary to motivate, justify or understand the premises upon which our work rests**. Just to clear up any possible further confusion: in the first four paragraphs of our Introduction, we cite 49 studies; of these, only 14 are purely theoretical/computational (29%); the remaining 35 are empirical (71%).

Thus, **depicting our work as if it was a computational, modeling study “more suited to a specialized computational audience” is a plain, complete misrepresentation of what we did at every level: background/motivation, results and interpretation**.

Study of mechanisms

I also fail to see their claims for “significant progress towards understanding the effects of a broad class of mechanisms”. The only real effort to address mechanisms comes through their added conceptual model in which they suggest the relative contribution of adaptation and leaky integration to both stimulus-dependent and intrinsic timescales. The ideas proposed are indeed interesting- especially the non-intuitive relationship between adaptation and intrinsic timescales which aren’t apparent from the predictions in Figure 1. However, they are relegated to the supplement, only integrated in the discussion, and presented with little clarity. Thus, these details are not likely to be appreciated by a general audience. Moreover, there is no effort to determine whether these features actually support the observations.

The reviewer criticizes here a statement we made in our reply to their first set of comments (not in the manuscript). The purpose of that statement was to clarify what our study is about – and we maintain that that statement is factually correct. In fact, although our work does not directly investigate the mechanisms underlying the temporal scales of stimulus-dependent and intrinsic processes, it sheds new light on the effects of such mechanisms at the level of neuronal representations and investigates the functional consequences that these mechanisms have in terms of neuronal coding. This is because **our work is not an empirical study of neuronal circuits and mechanisms – it is a phenomenological and functional study of poorly understood and poorly investigated neurophysiological processes, which also employs computational modeling to explore plausible, underlying mechanisms**. Here, we appositely and proudly use the term “phenomenological” that the reviewer has instead employed to diminish the impact and scope of our findings. We do so, because **we fully reject the notion that phenomenological studies are intrinsically inferior**, of lower interest and narrower scope or impact as compared to investigations of biological mechanisms – and, as such, they should be relegated to minor, specialistic journals and audiences. **Phenomenological studies, especially those exploring the functional implications of the phenomena under exam, have been and still are the backbone of systems neuroscience**. This is because one cannot even start investigating mechanisms unless the phenomena and their implications for cognition are explored, characterized and understood, and unless such understanding leads to some testable hypotheses about the mechanisms themselves, often with the support of computational modeling. We could mention many important, often seminal studies that are purely phenomenological in brain sciences, but the most recent and obvious one to cite here is the work of Siegle and colleagues of the Allen Institute, from which the mouse data analyzed in our study were taken. When we cited that study in our manuscript it was a preprint in Biorxiv; a few weeks ago it was published in a top-tier journal (*Nature*) – as many other, purely phenomenological neuroscience studies have been in recent months and years. But the best recent example of the deep understanding achieved by phenomenological studies is the Nobel Prize awarded in 2014 to John O’Keefe, May-Britt Moser and Edvard Moser for their discoveries of place and grid cells (it goes without saying that we are not

mentioning this to compare our work to theirs, but merely to point out the importance of phenomenological work).

Having clarified what our study is about, we hope it is also clear that **the computational model included in our revision is a bonus, a first attempt at exploring possible circuit mechanisms, without any pretense to be exhaustive and systematic.** This is why the model is never mentioned in the Results (there is no need for that), but only cited in the Introduction to motivate our approach and then in the Discussion to help explore mechanistic interpretations of our results on a more quantitative ground. And this is why all of the modeling work is shown in a supplementary figure – because **it is not critical to any of the conclusions of our study.** The modeling, being an initial attempt at exploring computational mechanisms, could certainly be expanded, but this would clearly go beyond the scope of our current study, which is not even remotely (we repeat it) a simulation-based, modeling work – it is an empirical investigation of brain processes. An expansion of the modeling work would require developing a dedicated, additional study. On the other hand, the modeling part could be removed from our current study without undermining any of the core conclusions derived from our experiments and data analyses. We are indeed very open to do so, if the editor and/or the reviewer think that this will simplify our narrative. What **we find unacceptable is to contest the validity, solidity and relevance of our study, based on the (intentional) lack of systematicity of the modeling work, which is included in our manuscript very explicitly as a marginal addition** (again, never cited as a result of the study) to the core body of empirical evidence we gathered and analyzed.

Receptive field size

Further, there is no effort to determine the contribution of changes in spatial integration, or the circuits that control spatial integration, to the observed temporal integration. Spatial integration is known to be differentially regulated in aroused/anesthetized conditions as compared to quiet wakefulness, and thus is a likely candidate to explain some of the observations. Indeed, it is concerning that the effect is only present when receptive fields might be of their larger sizes (as in the anesthetized/locomoting conditions), suggesting that much of the non-intrinsic effects could be explained by differences in receptive field size across the hierarchy. However, it would potentially be interesting if the intrinsic mechanisms might also be modulated by these same circuit mechanisms. However, this is not addressed.

Here the reviewer suggests a possible interpretation of our results of the quiet vs. active wakefulness comparison based on the enlargement of RFs in the latter state. **This is certainly an interesting observation that is worth mentioning as a discussion/interpretation point.** In fact, we already mentioned spatial integration as one of the precesses being modulated by the level of wakefulness. We have now expanded on this point in our **revised Discussion (p. 23/24)** to make more explicit that a common cause may possibly underlie the weakening of a variety of visual processes during quiet wakefulness. But why should it be “concerning that the effect is only present when receptive fields might be of their larger sizes (as in the anesthetized/locomoting conditions)”? **It is not clear how this would in any way invalidate any of our results, analyses, or conclusions or make them less interesting.** And why should we have determined (in this study) “the contribution of changes in spatial integration, or the circuits that control spatial integration, to the observed temporal integration”? We did not even know, before completing our study, if any difference of temporal integration would have been observed across the rodent visual system. Then why to focus on the enlargement of RFs, among the many processing properties that have been found to be modulated by the wakefulness state of rodents? Other properties could have been equally good candidates, such as the relationship between temporal integration and sparseness or reproducibility of firing rates. An answer to these questions is the same already provided above. **It is hard to investigate mechanisms of neurophysiological processes that are still**

unknown and not yet explored at the phenomenological and functional level. Once our results will have been published, they will likely be the starting point for further investigations, also at the mechanistic and circuitry level, possibly including the relationship between spatial and temporal integration postulated by the reviewer. But **we cannot follow up on all possible research directions opened up by our findings in a single study** – especially in a study that is already very rich in terms of datasets and quite complex and innovative in terms of analytical approaches. And especially after the major effort we were asked to carry out by adding awake recordings to our data pool. Similarly, **we cannot get started on a new major experimental/analytical tour-de-force at every round of revision**, based on new ideas, interpretations and insights that our current results can trigger on the reviewers, in a never-ending loop.

Conclusion

Thus, while the authors have potentially interesting results and ideas, it is not clear what mechanistic insights the authors have actually provided.

Given all the considerations above, I think that our answer to this criticism should be clear by now. **Our study, by design, does not provide new empirical evidence at the mechanistic level and we do not claim to do so anywhere in our manuscript. What our study does is to provide novel empirical evidence about neurophysiological phenomena that are poorly understood and poorly investigated, despite being critical to understand visual perception of dynamic scenes.** We characterize these phenomena (response and intrinsic time scales) within the same neuronal populations, across multiple visual areas that are organized hierarchically. We do so in two different rodent species, using three different datasets, comparing two visual hierarchies, and comparing processing under two wakefulness states and the anesthetized state. We investigate their functional role in mediating perceptual constancy of dynamic stimuli using innovative decoding analyses, and we explore the extent to which the two temporal processes (stimulus-driven and intrinsic) synergistically contribute to stabilize visual perception. Finally, we suggest computational mechanisms that may underlie the observed phenomena using modeling, thus paving the way for future empirical investigations of such hypothesized mechanisms. **We hope this makes it clear what our study does and the many ways our study is truly rich, informative and innovative** (despite not being mechanistic). Just to make an example: this is the only study to our knowledge that compares visual processing in partially overlapping visual pathways of rats and mice, thus providing conclusions with a general, across-species validity.

Third concern

The authors have reasonably addressed my concerns about the fits. However, I still find it a strange practice to lump all of the fits together given that the temporal integration of an exponential and a damped oscillation are fundamentally different, with the damped oscillation having a short initial window followed by a prolonged correlation. This is especially true given that the authors model suggests that the damped oscillation might be a key signature of the network state in their conceptual model.

As we have explained extensively in our previous revision, the method we apply is consistent and rigorous. We are glad to acknowledge that the reviewer is satisfied with our explanations and considers their concern addressed.

Fourth concern

The authors have improved the clarity of their abstract and introduction. However, the manuscript remains incredibly difficult to read. It suffers simultaneously from a lack of detail and clarity for how specific analyses were validated as well as an abundance of explanation for why certain routes were not taken. This made it difficult to navigate and determine which were the most important findings and how they contribute to the novelty and overall message of the manuscript.

The narrative in our manuscript has certainly grown in complexity because of the new analyses that were performed, the new datasets that were included and the new extended explanations that were required to motivate the premises of our study and interpret the results. In other words, the manuscript has grown in size and complexity because of our attempt at addressing all the concerns of the reviewers in the most careful and systematic way. Following the indications of both the reviewers, we have trimmed parts of our text, especially in the Discussion. We can certainly compress and streamline some parts of the narrative further, perhaps placing in the supplementary material some analyses/figures that are partially redundant. For example, we could keep in the main article the current Fig. 6, which most directly assesses the perceptual stability of the representations in conditions (single-object movie segments) where invariance is expected to play (as it does) a more substantial role in stabilizing the neuronal code, while relegating the sister analysis (currently shown in Fig. 5) to the supplementary material. In summary, we are open to work on our manuscript to make it way more compact and easier to read, depending on what the Editor will deem better.

Minor concerns

Why do all of the linear regressions have exactly the same slope? This seems unlikely from chance, and suggests that there is some constraint on the fit that biases towards having unique intercepts. It looks reasonable for data in Figure 4, but the V1 data in figure 5C have shorter timescales of discriminability for movies with short timescales, but similar timescales for longer timescale movies, and thus should have a steeper slope for the regression.

Indeed, as specified in the text (Results section, where a “common slope” is reported), the slope for the regression in Figure 5c is constrained to be exactly the same for all four areas. This is done for consistency with the methodology used for Figure 4, and to facilitate the interpretation of regression results, as one can focus only on the intercept to analyze inter-area differences. On the other hand, while the description given by the reviewer is not accurate (the timescales in V1 are not necessarily “similar” for longer-timescale movies, especially if one compares V1 and LL), it is true that generally speaking the quality of the linear fit is not as good as those shown in Figure 4. This is mostly driven by the two very strong LL outliers - the response discriminability timescale in LL is *much* longer than in the other areas for two of the movies. While exploiting this fact would have put us at an advantage in supporting our core findings, we decided to err on the side of caution and perform a *robust* linear regression (as explained in the Methods). This allowed us to greatly limit the impact of those two LL outliers, while preserving the simplicity and interpretability of the linear functional form used in the regressions of figure 4 (which, as documented in the methods, were performed by ordinary least squares).

Reviewers' Comments:

Reviewer #1:

Remarks to the Author:

I liked this paper after the first revision and have not changed my mind after the second revision. It is a valuable, thoughtful contribution to the literature which (if published) will be widely cited. I still think that the same message could be delivered in crisper fashion. This could be achieved with greater economy of words.

Reviewer #3:

Remarks to the Author:

In this work Piasini et al. present data recorded during natural movies from rodent visual areas. They find in later areas of the visual hierarchy a longer timescale of temporal correlation of population responses.

The consistency between the Allen mouse data and the data collected by the authors is nice and confirms their result in quite different experimental conditions (ie: mouse, awake, sometimes running; this overall point also made by R1).

I agree with many of reviewer 2's remaining concerns, but I see them primarily as issues of interpretation, not of results, or of data.

However, there is one essential issue that must be addressed. I concur with reviewer 2 that invariance is an interpretation based on their correlation data, not a result that has been firmly established. In fact, the Discussion now describes data from the Allen dataset showing increases up the hierarchy in the duration of responses to a flash. This suggests that the increases in temporal correlation length may not be about invariance, but instead about different temporal filtering. This would explain both the correlation plots and the decoding analyses because it is frame identity, that is, time, that is decoded. And I agree with R2 that their linear model does not completely rule this out. To fairly describe their results to readers, the authors must give this other interpretation early in the paper. This alternative temporal filtering possibility must be stated in the abstract and discussed in the Introduction.

I would also suggest softening the "invariance" interpretation throughout the paper, but this is just a recommendation. The way invariance is used in this work is related to, but not identical to, the way it's often used in object recognition. This is largely due to differences in "temporal invariance" vs object transformation invariance. Maybe these are totally resolvable, but it's not clear to me and I'm not sure it's key to this work.

Also, the title says "Intrinsic dynamics enhance...". There is a causal implication here that is again an interpretation, not a result, and is not strongly supported by data. Suggest title revision to "...reflected in intrinsic dynamics" or similar.

Re: feedforward vs intrinsic contributions: the true result here is about the PSTH correlation structure vs trial-to-trial correlation. Here too, these results are interesting by themselves. However, again, the interpretation of "intrinsic" is somewhat controversial as the authors don't have direct circuit measurements to show that what they measure is due to intrinsic/recurrent dynamics. A different set of experiments (e.g. using causal manipulations) might put this on firmer ground, but I see that as outside the current scope. I think readers in the field will see what analysis they did and understand the implications, so I think the use of "intrinsic" is fine.

Overall, I am enthusiastic about the results, which are strengthened by the mirroring of their effects in the Allen institute data. I am less enthusiastic about aspects of their interpretation. To me, text

revision around the alternative temporal filtering possibility seems essential. With that accomplished, I think this would be a solid contribution, whose results are novel, of appropriate scope, and appropriate for a broad audience.

Response to Reviewers - NCOMMS-19-37941B-Z

Reviewer #1 (Remarks to the Author):

I liked this paper after the first revision and have not changed my mind after the second revision. It is a valuable, thoughtful contribution to the literature which (if published) will be widely cited. I still think that the same message could be delivered in crisper fashion. This could be achieved with greater economy of words.

We thank the reviewer for their support. Following their advice, we have shortened the paper from about 60k characters to about 55k characters.

Reviewer #4 (Remarks to the Author):

In this work Piasini et al. present data recorded during natural movies from rodent visual areas. They find in later areas of the visual hierarchy a longer timescale of temporal correlation of population responses.

The consistency between the Allen mouse data and the data collected by the authors is nice and confirms their result in quite different experimental conditions (ie: mouse, awake, sometimes running; this overall point also made by R1).

I agree with many of reviewer 2's remaining concerns, but I see them primarily as issues of interpretation, not of results, or of data.

We thank the reviewer for their positive assessment.

However, there is one essential issue that must be addressed. I concur with reviewer 2 that invariance is an interpretation based on their correlation data, not a result that has been firmly established. In fact, the Discussion now describes data from the Allen dataset showing increases up the hierarchy in the duration of responses to a flash. This suggests that the increases in temporal correlation length may not be about invariance, but instead about different temporal filtering. This would explain both the correlation plots and the decoding analyses because it is frame identity, that is, time, that is decoded. And I agree with R2 that their linear model does not completely rule this out. To fairly describe their results to readers, the authors must give this other interpretation early in the paper. This alternative temporal filtering possibility must be stated in the abstract and discussed in the Introduction.

I would also suggest softening the "invariance" interpretation throughout the paper, but this is just a recommendation. The way invariance is used in this work is related to, but not identical to, the way it's often used in object recognition. This is largely due to differences in "temporal invariance" vs object

transformation invariance. Maybe these are totally resolvable, but it's not clear to me and I'm not sure it's key to this work.

We agree that giving the temporal filtering interpretation more prominence and softening the invariance interpretation would result in a clearer presentation of the factors that could possibly contribute to difference in timescales between areas. Following the reviewer's recommendation, we now present the temporal filtering hypothesis directly in the abstract and in the introduction. We have also made changes throughout the text in order to soften the invariance interpretation and to make the temporal filtering hypothesis more prominent. This involved changes in 9 paragraphs, uniformly distributed between the relevant parts of the text (1 in the Abstract, 4 in the Introduction, and 4 in the discussion). We have highlighted these changes in blue.

Also, the title says "Intrinsic dynamics enhance...". There is a causal implication here that is again an interpretation, not a result, and is not strongly supported by data. Suggest title revision to "... reflected in intrinsic dynamics" or similar.

We thank the reviewer for pointing this out. We have gone even further than what the reviewer suggested, by removing entirely the mention of intrinsic dynamics from the title. The title now reads "Temporal stability of stimulus representation increases along rodent visual cortical hierarchies".

Re: feedforward vs intrinsic contributions: the true result here is about the PSTH correlation structure vs trial-to-trial correlation. Here too, these results are interesting by themselves. However, again, the interpretation of "intrinsic" is somewhat controversial as the authors don't have direct circuit measurements to show that what they measure is due to intrinsic/recurrent dynamics. A different set of experiments (e.g. using causal manipulations) might put this on firmer ground, but I see that as outside the current scope. I think readers in the field will see what analysis they did and understand the implications, so I think the use of "intrinsic" is fine.

We are glad to see that the reviewer agrees that our usage of the word "intrinsic" is appropriate when referring to feedforward vs intrinsic contributions.

Overall, I am enthusiastic about the results, which are strengthened by the mirroring of their effects in the Allen institute data. I am less enthusiastic about aspects of their interpretation. To me, text revision around the alternative temporal filtering possibility seems essential. With that accomplished, I think this would be a solid contribution, whose results are novel, of appropriate scope, and appropriate for a broad audience.

We thank the reviewer for their enthusiastic support of our results, and for the precious feedback provided about the interpretation of our results. As detailed in the responses to the previous points, we have implemented all suggested changes, and we believe our paper is all the better for it.